# CO$_2$ and CH$_4$ dynamics in a eutrophic tropical Andean reservoir

**Eliana Bohórquez-Bedoya**[1,2], **Juan Gabriel León-Hernández**[3], **Andreas Lorke**[2]*, **Andrés Gómez-Giraldo**[1]

**1** Department of Geosciences and Environment, Universidad Nacional de Colombia, Medellín, Antioquia, Colombia, **2** Institute for Environmental Sciences, University of Kaiserslautern-Landau, Landau, Rhineland-Palatinate, Germany, **3** Universidad Nacional de Colombia Department of Engineering, Palmira, Valle del Cauca, Colombia

* a.lorke@rptu.de

**Data Availability Statement:** The datasets generated during and/or analyzed during the current study are available in the Figshare repository, 736 https://doi.org/10.6084/m9.figshare.20288547.v1 (Bohórquez et al., 2022).

## Abstract

We studied the dynamics of methane (CH$_4$) and carbon dioxide (CO$_2$) in a eutrophic tropical reservoir located in the Colombian Andes. Temporal and spatial dynamics were addressed through sampling during six field campaigns conducted throughout a two-year period. We monitored fluxes at the air-water interface, dissolved gas concentrations, physical and chemical properties of the water column, microstructure profiles of turbulence, and meteorological conditions. Throughout the study period, the reservoir was a persistent source of CH$_4$ to the atmosphere with higher emissions occurring in the near inflow region. During periods of low water levels, both the emissions and surface concentrations of CH$_4$ were higher and more spatially heterogeneous. The measured CO$_2$ fluxes at the air-water interface changed direction depending on the time and location, showing alternating uptake and emissions by the water surface. Mass balances of dissolved CH$_4$ in the surface mixed layer revealed that biochemical reactions and gas evasion were the most significant processes influencing the dynamics of dissolved CH$_4$, and provided new evidence of possible oxic methane production. Our results also suggest that surface CH$_4$ concentrations are higher under more eutrophic conditions, which varied both spatially and temporally.

## Introduction

Freshwater reservoirs emit globally significant amounts of greenhouse gases (GHG), including carbon dioxide (CO$_2$) and methane (CH$_4$), to the atmosphere. Estimates suggest that GHG emissions from reservoirs may account for approximately 1.3% to 7% of the total anthropogenic emissions in CO$_2$ equivalents [1,2]. Tropical inland waters are recognized as particularly strong sources because of the high content of organic matter and favorable temperature for microbial activity all year round [3–6].

In freshwater reservoirs, CO$_2$ and CH$_4$ can be produced and consumed through biochemical processes occurring in the sediment and in the water column, and surface fluxes are additionally affected by vertical transport processes in the sediment, the basin interior and at the water surface. Turbulent mixing produced by wind stress [7,8], convective mixing caused by

**Funding:** This work was founded by Ministerio de Ciencia Tecnología e Innovación de Colombia - MinCiencias, Call 714-2015 "Convocatoria Investigación y Desarrollo Tecnológico e Innovación en Ambiente, Océanos y Biodiversidad". https://minciencias.gov.co/ EBB has been funded by Ministerio de Ciencia Tecnología e Innovación de Colombia - MinCiencias, Call 757-2016 "Doctorados Nacionales" The funders had no role in study design, data collection and analysis, decision to publish, or preparation of the manuscript.

**Competing interests:** The authors have declared that no competing interests exist.

surface cooling [9], and advective transport [10] are the main physical processes involved in transport and evasion of dissolved gases from reservoirs.

Reservoirs are typically characterized by a longitudinal gradient from the riverine inflow region to the dam, as well as by high temporal variability of hydrodynamic transport processes in response to meteorological forcing at seasonal and diurnal scales. Primary production is also a major factor controlling the dynamics of CO$_2$ and CH$_4$ in the surface layer of aquatic systems [11–15]. Thus, the interplay of dynamic physical drivers with production/consumption processes regulate the complex spatial patterns and dynamics of atmospheric emissions. Mechanistic mass balance approaches of dissolved greenhouse gases in the surface layer of a reservoir can be approximated in terms of the horizontal advection, vertical fluxes, production/consumption of gases in the surface layer, and atmospheric evasion rates [7,16–18].

Most existing studies on GHG emissions from tropical reservoirs have focused on lowland or plain regions in South America, such as the Amazon forest and savannas [19–22], whereas the emissions from reservoirs located in different biomes, such as the Andean mountains, remain largely unexplored. Moreover, there has been a scarcity in mechanistic approaches and understanding of the processes governing gas flux dynamics and their seasonal and diel variations in these reservoirs. The objective of this study is to analyze the dynamics of CO$_2$ and CH$_4$ concentrations and emissions in a eutrophic tropical Andean reservoir. We hypothesized that the dynamics of greenhouse gases within a eutrophic reservoir are subject to seasonal variations in response to changes in hydrological conditions, and that the spatial distribution of these gases varies with proximity to the river inflow.

## Materials and methods

### Study site

Porce III is a canyon type long and narrow hydroelectric reservoir flooded in 2010, with a surface area of 4.61 km$^2$, 12 km length, and average and maximum water depths of 45 m 137 m, respectively (Fig 1A). It is located at 690 m.a.s.l. in a wet tropical forest of the Colombian Andes, and is part of a series of cascading reservoirs, with two upstream reservoirs (Troneras and Porce II). As expected from the morphology of the reservoir, the predominant direction of wind speed follows the canyon shape, and the main direction of the wind is from the north-east to the south-west in the morning and in the opposite direction in the afternoon.

The main tributary of Porce III reservoir is the Porce river, which transports municipal wastewater from a metropolitan area with almost 4 million inhabitants and multiple land uses, including urban, agricultural and industrial. The river is dammed upstream in the Porce II reservoir and reaches Porce III with significant concentrations of nitrate, ammonium and phosphate (S1 Table). These inflow characteristics promote eutrophic conditions in both Porce II and Porce III reservoirs, as does the consistently high atmospheric and water temperature of around 25˚C (Table 1), which favors algae growth. The trophic state index (TSI) (Toledo et al., 1983) estimated from the monitoring conducted by the reservoir operator, indicates constantly eutrophic condition of Porce III reservoir between 2016 and 2018 (TSI >54) (S7 Fig).

Three sampling stations along the reservoir were selected to evaluate the longitudinal variability and were considered as representative of three zones of the reservoir (station name and mean depth during the study period are provided in parentheses): inflow zone (P3, 24.3 m), mid-lake zone (P2, 53.0 m) and dam zone (P1, 112 m) (Fig 1A).

### Data collection

**Sampling and data overview.** To identify seasonal variation among the campaigns, hydrological and meteorological data were continuously collected from May 2017 to February

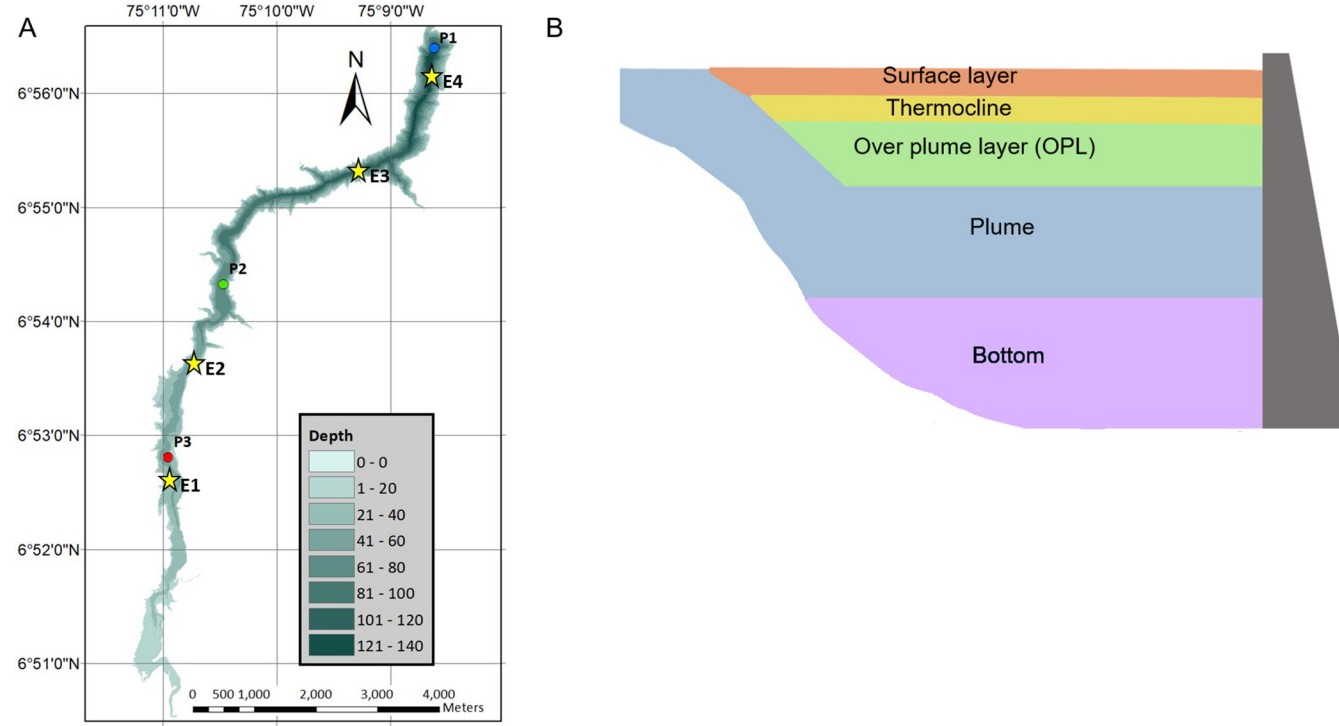

**Fig 1. Porce III reservoir coordinates.** A) bathymetry and sampling stations for this study (circles with the letter P and for the rutinary water quality monitoring (stars and letter E). B) Scheme of a typical vertical structure along a longitudinal cross section of Porce III reservoir. Colors and labels mark layers with distinct physical and physicochemical characteristics.

2019. Six field campaigns were carried out, the first in May 2017 and the remaining five between May 2018 and February 2019. The campaigns included periods of high (> 673 m.a.s. l.), low (< 665 m.a.s.l.) and medium (665 to 673 m.a.s.l.) water level. The field campaigns were named according to the basin's hydrological condition and the water level during the sampling period. Thus, the campaigns were named: high-level-wet C1-H-Wet (May/2017); high-level-wet C2-H-Wet (May/2018); low-level-Dry C3-L-dry (Aug/2018); low-level-dry-wet-transition C4-L-DWT (Sep/2018); medium-level-Wet C5-M-Wet (Nov/2018) and medium-level-Dry C6-M-Dry (Feb/2019) (Fig 2, S2 Table). Two field campaigns were affected by El Niño that is associated with drier weather in the study region. The final campaign (C6-M-Dry) took place during a period of drought after several months of El Niño conditions, when the operation of the cascading reservoirs exhibited unusual patterns. The discharge at upstream and Porce III

**Table 1. Meteorological variables during the study period from 09-May-2017 to 28-Feb-2019.**

| Variable | Units | min | max | mean ± std | | |
|---|---|---|---|---|---|---|
| Surface water temperature | ˚C | 22.4 | 28.7 | 25.0 | ± | 0.83 |
| Air temperature | ˚C | 19.6 | 33.0 | 23.7 | ± | 1.91 |
| Relative humidity | % | 50.5 | 98.3 | 88.6 | ± | 7.26 |
| Atmospheric pressure | hPa | 1001 | 1011 | 1006 | ± | 1.7 |
| Cloud cover | ----- | 0.00 | 1.00 | 0.88 | ± | 0.20 |
| Wind speed ($U_{10}$) | m s⁻¹ | 0.00 | 13.0 | 3.7 | ± | 1.4 |

Minimum (min), maximum (max) and mean ± standard deviation (std).

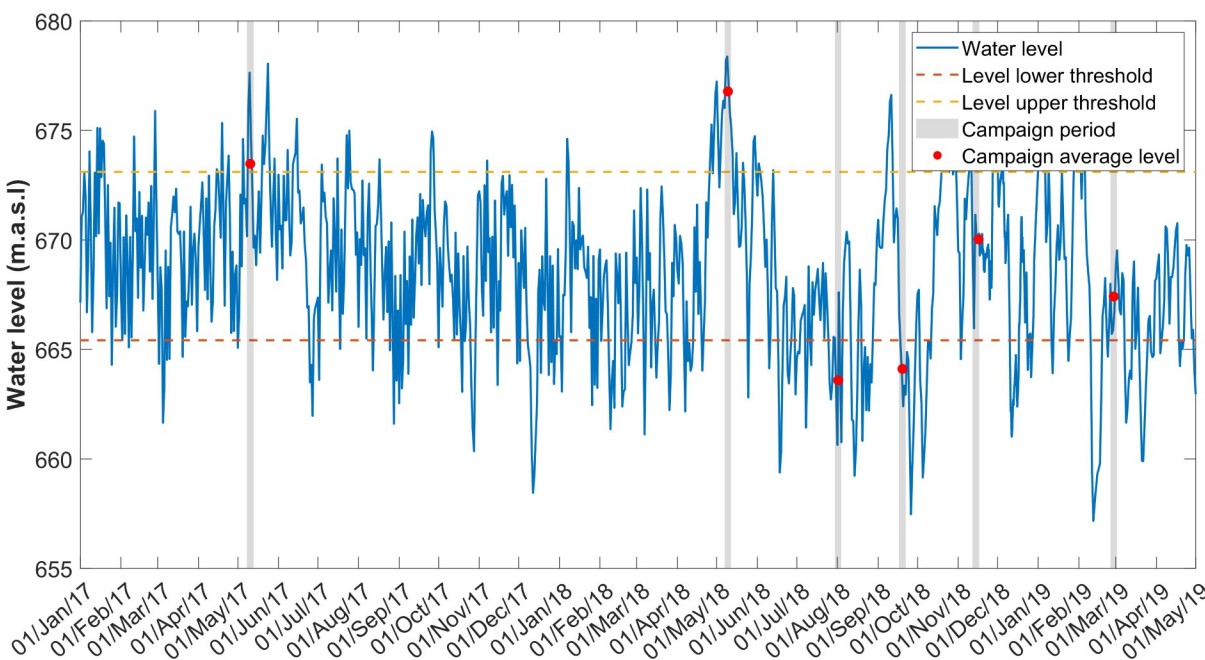

**Fig 2. Water level (1 h resolution).** Campaigns were conducted at a mean water level above the upper threshold (673.1 m.a.s.l., red horizontal line), below the lower threshold (665.4 m.a.s.l., blue horizontal line) or between both thresholds were classified as high level (H), low level (L) or medium level (M), respectively.

reservoirs were abruptly opened and closed a few days prior to this campaign, and these changes were clearly reflected in the water level (Fig 2).

To study daily patterns, we conducted sampling at the dam zone (sampling point P1) three times a day during: morning (P1-M, 08:00–12:00), afternoon (P1-A, 14:00–20:00), and night (P1-N, 22:00–03:00). Typically, sampling followed a sequence: inflow zone P3 during the daytime, mid-lake zone P2 during the daytime, and finally, the dam zone (P1), capturing diurnal dynamics in all field campaigns, except the initial one (C1-H-Wet). In the study area, the daylight hours are very stable from 06:00 to 18:00 throughout the year, so some samples in the afternoon were taken during the first nighttime hours. Throughout most of the campaigns, surface measurements were conducted twice a day at the mid-lake and inflow zones P2 and P3. For spatial analysis, we opted for comparing the morning samples at the dam zone (P1-M) with the mid-lake and inflow zones, which were primarily monitored in the morning and during midday.

To characterize the vertical structure of the reservoir and its variations both horizontally and throughout the study period, we measured vertical profiles of physicochemical variables. The Trophic State Index (TSI), based on Toledo (1984), was provided by the reservoir operator based on its periodic water quality monitoring. In situ samples were collected to measure dissolved gas concentrations (CO$_2$ and CH$_4$) and water-air gas fluxes, enabling estimation of the gas transfer velocity. Ultimately, these results were utilized to estimate CH$_4$ budgets in the surface layer using a mechanistic mass balance approach.

## Meteorological and hydrological data collection

A weather station (Davis Vantage Pro2) was installed floating on the water surface to monitor meteorological variables, including wind speed (at approximately 2.0 meters above the water surface), air temperature, relative humidity, atmospheric pressure, and solar radiation, at

hourly intervals. Two additional stations were available to fill in gaps during periods of missing data (S1A Appendix). Surface water temperature was monitored with hourly resolution using a Hobo thermistor located at 0.2 m depth at the dam zone (P1). Wind speed was corrected at 10 m height based on [23]. For determining the hydrological state of the reservoir during the study period, water level, inflows and outflows were provided by the operator of the reservoir at 1 hour resolution (Empresas Públicas de Medellín—EPM).

## Physical and chemical data and vertical structure

Vertical profiles of water temperature and turbidity were measured using a conductivity-temperature-depth probe (CTD, SeaBird SB25) and profiles of dissolved oxygen concentration (DO) and pH were measured using a multiparameter water quality probe (MWQS, YSI EXO 1). In addition, vertical profiles were measured using a turbulence probe (MicroCTD of Rockland Scientific) with a minimum of three casts at each sampling station (P1, P2, P3). The probe measures turbulent fluctuations of horizontal flow velocity and water temperature at a sampling rate of 512 Hz.

Vertical layering was defined from measured profiles before sampling of CO$_2$ and CH$_4$ concentrations in each layer. The vertical structure of the reservoir was generally stable over time and divided into five layers from top to bottom: Surface layer, thermocline, over-plume layer (OPL), inflow plume, and bottom. The spatial variations of the layered vertical structure are illustrated in Fig 1. The currents produced by selective withdrawal at the dam were always located in the layer defined by the plume.

Secchi disk depth was measured at least once at each sampling station during the diurnal sampling, except for the low-level-dry-wet-transition campaign C4-L-DWT. Secchi depth served as an indicator of the trophic state, and while it was not directly measured, elevated algae concentrations and the presence of green-colored water at the surface were consistently evident throughout the study period.

## Dissolved CO$_2$ and CH$_4$ in the water columna

Concentration of dissolved CO$_2$ and CH$_4$ were measured in duplicates at each sampling station (P1, P2, P3) by the headspace technique followed by GC-FID analysis [24]. Samples were taken at the water surface and in the other layers and immediately transferred to 30 mL vials, crimp-closed with butyl septa and poisoned with HgCL$_2$ (1 mg L$^{-1}$). In the lab, a 20 mL headspace was created by injecting N$_2$ with a syringe and needle through the septa while keeping the bottle bottom-up, and simultaneous withdrawal of 20 mL of water through a second needle. The liquid phase was equilibrated with the gas phase by shaking each vial for 2 min and left to equilibrate at ambient temperature for more than 1 h. The gas phase was then analyzed in a gas chromatograph (Shimadzu GC-2014 equipped with a flame ionization detector). A commercial standard at 1000 ppmv and custom-made mixtures of nitrogen and pure methane were used for calibration. The dissolved gas concentrations under in-situ conditions were calculated using the temperature dependent solubility coefficients provided by [25].

## Vertical gas diffusion in the water column

To estimate the vertical diffusion of dissolved gases in the water column, vertical profiles of turbulent diffusivity ($K_z$) were calculated following Osborn (1980) (Eq (1)):

$$K_z = \gamma_{mix} \frac{\epsilon_m}{N^2} \tag{1}$$

where $\gamma_{mix}$ is the mixing efficiency estimated according to the turbulent intensity parameter

dependent mixing regime [26]; $\epsilon_m$ is the dissipation rate of turbulent kinetic energy; $N$ is the buoyancy frequency ($N^2 = (g/\rho)(\partial\rho/\partial z)$, where g is the gravitational acceleration, $\rho$ is the temperature dependent water density and $z$ is water depth), which was calculated from the CTD temperature profiles.

Vertical profiles of dissipation rates of turbulent kinetic energy ($\epsilon_m$) were estimated by processing the data from two shear sensors and one micro-temperature sensors of the microstructure profiler (MicroCTD). Data quality verification and the calculation of $\epsilon_m$ for each profile based on the Nasmyth shear spectrum (Oakey, 1982) were performed using manufacturer-provided scripts (ODAS v4.4.04 MATLAB library provided by Rockland Scientific). Vertical profiles of $\epsilon_m$ were derived by averaging the profiles from both shear probes. Subsequently, dissipation rates were obtained at 0.50 m intervals in each profile through cubic interpolation. Finally, a representative profile of $\epsilon_m$ was estimated by logarithmically averaging over 2-m bins.

## Emissions of CO$_2$ and CH$_4$, and $k_{600}$

Fluxes of CO$_2$ and CH$_4$ across the air-water interface were measured using floating chambers. Two plastic chambers (volume 40 L, surface 0.15 m$^2$), each equipped with a rubber stopper allowing for gas sampling with a syringe and needle, were deployed simultaneously from a boat. Each deployment lasted for 45 min and gas samples were collected in 15 min intervals. After collection, the gas samples for CO$_2$ analysis were immediately stored in 20 mL pre-evacuated vials. For CH$_4$ analysis, the vials were prepared with a KCl solution and were kept bottom-up while injecting the sample and simultaneously withdrawing 10 mL of water through a second needle. Subsequently, gas concentrations in the samples were analyzed using gas chromatography (Shimadzu GC-2014 equipped with a methanizer and a flame ionization detector). Water samples for measuring dissolved gas concentration in surface water ($C_w$) were collected at 0.20 m depth the during chamber deployments, additional to the ones collected for the concentration profiles, and analyzed using the method explained above.

Fluxes across the air-water interface, $F_{g,T}$, were calculated using linear regressions based on the concentration change of the gas (CO$_2$ or CH$_4$) over the 45 min sampling period, the chamber volume ($V_{cham}$) and surface area ($A_{cham}$) as:

$$F_{g,T} = (dC_{cham}/dt_{cham})V_{cham}/A_{cham} \tag{2}$$

In the first place, we accepted measurements associated with regressions with a reasonably good coefficient of determination ($r^2 > 0.70$) and estimated gas fluxes of CO$_2$ and CH$_4$ by averaging over the replicated chamber deployments. Then, the gas transfer velocity was estimated as:

$$k_{g,T} = \frac{F_{g,T}}{(C_w - C_{eq})} \tag{3}$$

where $k_{g,T}$ is the gas transfer velocity and $C_w$ and $C_{eq}$ are the dissolved gas concentrations in the surface water and the concentration in equilibrium with the atmosphere, respectively. The gas-specific transfer velocities at in-situ temperature (Eq (3)) were then normalized to a Schmidt number of 600 (Sc = 600, for CO$_2$ at 20˚C) to obtained the normalized gas transfer velocity ($k_{600}$) (Jähne et al., 1987). CH$_4$ flux measurements that were affected by ebullition were discarded for the analysis of diffusive fluxes (see methodology in the S1B Appendix).

**Table 2. Description of the terms of the mass balances.**

| Term | Expression | Description |
|---|---|---|
| $V\Delta C/\Delta t$ | $\frac{\Delta C}{\Delta t}V$ | Rate of change of mass in the control volume ($V$). Estimated from observed changes of dissolved gas concentration ($C$) over periods ($\Delta t$) of several hours (1–8 h). It is positive for increasing concentration. |
| *diff_low* | $\frac{C_{OPL}-C_{SML}}{h}Kz_{Therm}A_s$ | Diffusive exchange at the base of the surface (SML). $C_{OPL}$ and $C_{SML}$ are the concentrations measured in the overplume layer (OPL) and the surface layer (SML) (closest to the thermocline), $h$ is the thickness of the thermocline and $K_{z\_Therm}$ is the vertical diffusivity averaged in the thermocline. Incoming fluxes to the control volume were considered as positive and outgoing fluxes as negative. $A_s$ is the surface area. Since turbulence measurements were not done during P1-M and P1-A, campaigns C5-M-Wet and all C6-M-Dry, we assumed $K_{z\_Therm}$ of those as an average of the successful measurements at the other sites. |
| evasion | $F_{g,T}A_s$ | Atmospheric exchange. The flux, $F_{g,T}$ was calculated using Eq (2) and surface area $A_s = 1$ m². The negative sign of the evasion term indicates a flux from water to air. |
| advection | $\frac{C_{SML\_i+1}-C_{SML\_i}}{\Delta x}U_{SML}V$ $U_{SML}(z) = U_w - \frac{u_*}{\kappa}\ln\left(\frac{z}{z_0}\right)$ | Horizontal advective transport. ($C_{SML\_i+1}$-$C_{SML\_i}$)/$\Delta x$ is the mean horizontal gradient of the concentration between two sampling sites separated by the distance $\Delta x$. Upstream and downstream concentrations were measured around one day apart. Integrated concentrations over the surface depth were used. $U_{SML}$ is the mean flow velocity in the surface layer, since overflow of the river plume was not observed during the study period. It was estimated by integrating the theoretical logarithmic profile produced by the wind speed $U_{10}$ over the surface layer, every 1 cm of depth. The control volume: $V$ (m³) = 1 m × 1 m × $h_{SML}$ (m) where $h_{SML}$ is the thickness of the surface layer. In using this approach, we assume that horizontal concentrations follow a spatial pattern that does not change significantly over an interval of 1–2 days (see S1C Appendix). |
| Reactions | To be solved. | This term represents the net production or consumption of dissolved $CH_4$ in the SML. |

## Surface mass balances and net methane production

Mass balances of dissolved $CH_4$ were estimated for the surface layer at the local scale of the sampling sites (an area of 1 m²). The rate of change of the mass of dissolved $CH_4$ in the control volume ($V\Delta C/\Delta t$) was considered to be caused by vertical turbulent transport across the thermocline at the lower base of the surface layer (*diff_low*), evasion to the atmosphere (*evasion*), horizontal advective transport (*advection*) and biochemical reactions (*reactions*) (Eq (4), Table 2):

$$V\Delta C/\Delta t = diff\_low - evasion + advection + reactions \qquad (4)$$

Eq (4) was solved for the *reactions* term, which could not be estimated from measurements. These results are equivalent to the net methane production ($P_{net,CH4}$ in µmol L$^{-1}$ d$^{-1}$, with consideration of unit conversion factors), which includes net production (positive rates) or consumption (negative rates) of $CH_4$ in the control volume by biochemical pathways.

## Results

### Meteorological conditions

The meteorological variables during the study period reflect typical conditions of the wet tropical forests in the Colombian Andes (Table 1): small seasonality in solar radiation, high water and air temperatures around 25.0 ± 0.83 and 23.7 ± 1.9°C, respectively, relative humidity higher than 50% and high cloud cover (88% on average with lower values in the dry season Dec 2018—Feb 2019). The maximum wind speed ($U_{10}$) during the study period was of 13.0 m s$^{-1}$ and the average of the time series was 3.7 ± 1.9 m s$^{-1}$, with a slight increase in the period Jun-Sep/2018 and a decrease in the period Nov-Dec/2019 (S1 Fig).

### Water column characterization

**Overview.** The stratification of the reservoir showed a persistent spatial pattern with distinct layers developing from the inflow zone (P3) towards the dam zone (P1). In the inflow region (~ 24 m water depth), there is a dominant inflow plume and small thermocline and

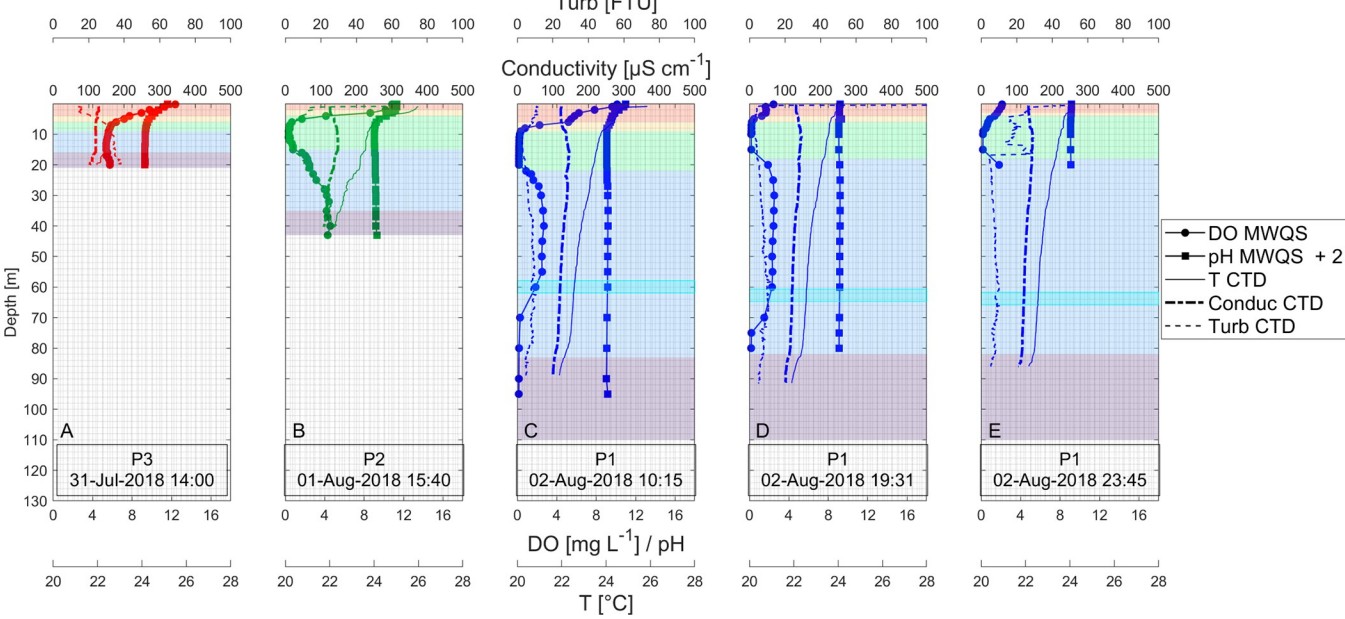

**Fig 3. Vertical profiles of physicochemical variables in Porce III reservoir during C3-L-Dry.** A) at the inflow zone (P3), B) at the middle zone (P2), C) at the dam zone in the morning (P1-M), D) at the dam zone in the afternoon (P1-A), E) at the dam zone in the night (P1-N). Each panel shows dissolved oxygen (DO), pH, water temperature (T), electrical conductivity (Conduc) and turbidity (Turb) measured with the multiparameter water quality sonde (EXO YSI, MWQS) or the CTD (SeaBird 25) as shown in the legend. The background color represents the layers according to the previously defined conventions (Fig 1B).

surface layer, with a thin bottom layer appearing only in a few cases. In the mid-lake zone (~ 53 m depth) the inflow plume became thicker, but the observed vertical structure from the inflow zone persisted. In the lacustrine dam zone (~ 112 m depth), an anoxic bottom layer developed, except during the medium-level-wet campaign in Nov/2018 (C5-M-wet), when the cold inflow plume propagated along the bottom up to the dam (S4 Fig).

The reservoir was weakly stratified during all field campaigns and characterized by five distinct and generally stable vertical layers, from the top to the bottom called surface layer, thermocline, over plume layer (OPL), the river plume characterized by being oxygenated and the anoxic bottom. During daytime, a diurnal thermocline developed (e.g. during the low-level-dry campaign C3-L-Dry in Figs 3 and S2). The analysis on the water column was important for define the vertical structure and for quantifying the vertical exchange of the surface layer.

**Temperature and stratification.** The continuous measurements at the dam zone (P1) showed that water surface temperature ranged from 22 to 29°C (mean 25.0 ± 0.97) during the study period. Slightly lower temperatures were observed in the rainy seasons (May and Nov/2018) and the highest temperatures were observed in the dry season (Aug/2018) (S1 Fig).

During the field campaigns water surface temperature ranged from 25 to 28°C. Below the thermocline, in the over-plume layer, water temperature ranged from 22 to 24°C. The thickness of the surface layer ranged from 1.5 to 8 m and was generally larger in the morning and smaller in the afternoon at the dam zone P1 and lower at the mid-lake zone P2 compared to the other two sampling stations. The thermocline was particularly thick during the high-level-wet campaign in May/2018 (C2-H-Wet), when it reached up to 13.5 m, whereas it ranged from 1 to 4.5 m thickness during the other campaigns (e.g. during the low-level-dry campaign C3-L-Dry in Figs 3 and S2).

**Dissolved oxygen, pH and Secchi disk depth.** Dissolved oxygen (DO) profiles showed that the inflow zone P3 was always under oxic conditions showing DO saturation > 44% even

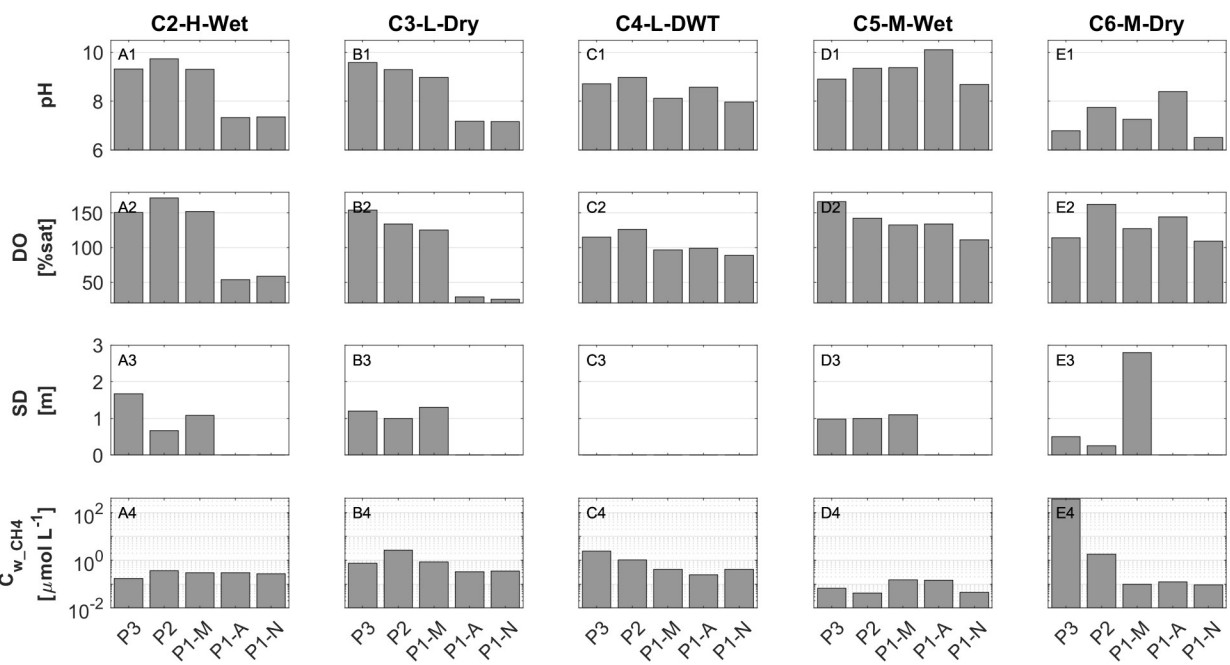

**Fig 4. Physicochemical characteristics of the surface layer depicted across various sampling campaigns (columns).** The panels are identified as follows: A) campaign C2-H-Wet, B) campaign C3-L-Dry, C) campaign C4-L-DWT, D) campaign C5-M-Wet, E) campaign C6-M-Dry, and 1) pH, 2) dissolved oxygen, 3) Secchi disk depth (SD), and 4) surface concentration of $CH_4$ ($C_{w\_CH4}$) on a logarithmic axis. Note: SD measurements were unavailable for campaign C4-L-DWT and at station P1-A during C2-H-Wet and C3-L-Dry due to darkness.

below the thermocline. In the mid-lake to dam zones (P2 to P1), we observed a rapid vertical decrease of DO, usually reaching hypoxic conditions (<10% DO saturation) below the thermocline (at ~7–8m) (e.g. during the low-level-dry campaign Figs 3 and S2).

The surface water was supersaturated with DO during all field campaigns and at all three sampling stations during daytimes (114–172%, Fig 4 –panels A2 to E2), indicating a continuous positive net production of the ecosystem. This is further supported by the alkaline pH values (8.1 to 9.7) found in the daytime samples and by the strong day-night differences observed in both DO and pH during the wet campaigns conducted between May and Nov/2018 (C2-H-Wet to C5-M-Wet). Especially throughout the campaigns conducted between May and Aug/2018—specifically, the high-level-wet campaign C2-H-Wet and the low-level-dry campaign C3-L-Dry, we observed diel changes in dissolved oxygen (DO) saturation and pH with amplitudes of about 100% and 2 units, respectively (Fig 4 –panels A1 to B2). These observations suggest a strong influence of photosynthetic activity on the diel dynamics of both variables. Subsequent field campaigns in Sep and Nov/2018, i.e. the low-level-dry-to-wet-transition campaign C4-L-DWT and the medium-level-wet campaign C5-M-Wet, showed slightly lower values of DO and pH, along with smaller day-night differences. In these campaigns, diel changes in DO saturation were 10% and 23%, while changes in pH were 0.6 and 1.4 units (Fig 4 –panels C1 to D2). During the final campaign in Feb/2019 (medium-level-dry campaign C6-M-Dry), relatively low surface pH values close to neutral conditions (ranging from 6.8 to 8.4) were observed during daytime. However, DO supersaturation persisted, and lower values of both variables were found at night, indicating the maintenance of productive conditions (Fig 4 –panels E1 and E2). Notably, in campaigns where afternoon samples (P1-A) were collected during daylight (Sep/2018 to Feb/2019), we observed significant increases in both DO and pH from morning to the afternoon (Fig 4 –panels C1 to E2).

In the campaigns conducted from May to Sep/2018, dissolved oxygen (DO) and pH levels were consistently lower in the dam zone P1 compared to the upstream sections of the reservoir (P3 and P2). Notably, the mid-lake zone P2 exhibited characteristics indicative of stronger photosynthetic activity throughout most of the field period, including the campaigns in May and Sep/2018, as well as Feb/2019 (the high-level-wet campaign C2-H-wet, low-level-dry-to-wet-transition campaign C4-L-DWT and the medium-level-dry campaign C6-M-Dry). During these campaigns, higher pH and DO levels were observed at P2 than at the other two sites. Furthermore, Secchi disk depth (SD) was often lower in the mid-lake zone P2 (Fig 4 –panels A3 to E3). Following an intense rainfall event (S4F Fig), significant spatial variations in SD were noted during the final campaign in Feb/2019 (C6-M-Dry), when SD was very low at P2 (0.67 m) and relatively clear surface water at P1 (2.8 m).

## Dissolved CO$_2$ and CH$_4$

The concentrations of dissolved CO$_2$ in surface water were generally high (379 ± 226 μmol L$^{-1}$) and always exceeded the atmospheric equilibrium concentrations. The emissions measured by the floating chambers, in contrast, showed that the flux direction was from air to the water (detailed below). We concluded that the alkaline pH of the surface water in the reservoir may have caused an overestimation of the CO$_2$ concentration by the headspace procedure (see S1D Appendix) and consequently we did not consider the measured CO$_2$ concentrations in the following analysis.

Methane (CH$_4$) concentrations typically increased in deeper layers, occasionally exhibiting peaks between the thermocline and the layer below (referred to as the over-plume layer or OPL) (e.g. during the low-level-dry-wet-transition campaign Figs 5 and S3). Concentrations decreased in the inflow plume where dissolved oxygen (DO) concentrations were slightly higher. In instances where the bottom layer was anoxic, dissolved CH$_4$ concentrations were notably high (∼ 5 × 10$^1$–5 × 10$^2$ μmol L$^{-1}$) below a depth of 80 m. Within the hypolimnetic over-plume layer OPL, CH$_4$ concentrations ranged from ∼ 1 to 5 μmol L$^{-1}$, with smaller

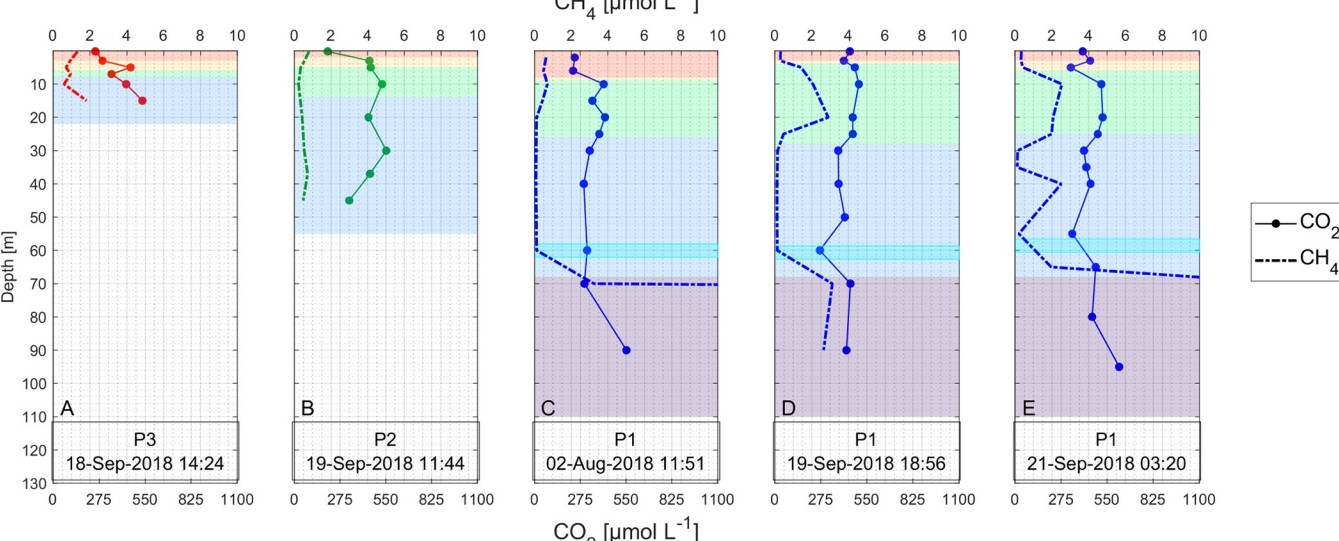

**Fig 5. Dissolved CO$_2$ and CH$_4$ profiles during C4-L-DWT.** The background color represents the layers according to the previously defined conventions. Note the out-of-scale CH4 concentrations at the lowest sampling depth at P1. The hidden values are: panel C)145 μmol L$^{-1}$ at 90 m depth, panel E) 41.4 and 441 μmol L$^{-1}$ at 95 m and 80 m depth, respectively.

**Table 3. Seasonal averages (mean ± std) of the variables related to the atmospheric fluxes of CH$_4$, surface water concentrations ($C_{w-CH4}$) excluding exceptional values ( > 69 μmol L$^{-1}$), measured diffusive atmospheric flux ($F_{CH4}$ and $F_{CO2}$).** Also, the ratios of mean values for dry, wet and dry-wet transitions (DWT) as well as low, medium and high-water level seasons are presented.

| Season | Campaign | $C_{w-CH4}$ (μmol L$^{-1}$) | $F_{CH4}$ (mmol m$^{-2}$ d$^{-1}$) | $F_{CO2}$ (mmol m$^{-2}$ d$^{-1}$) |
|---|---|---|---|---|
| High water level | C1-H-Wet C2-H-Wet | 0.46 ± 0.06 | 0.46 ± 0.6 | 14.3 ± 59 |
| Low water level | C3-L-Dry C4-L-DWT | 1.09 ± 0.92 | 1.68 ± 0.8 | 34.7 ± 97 |
| Medium water level | C5-M-Wet C6-M-Dry | 0.40 ± 0.88 | 0.73 ± 0.2 | -17.3 ± 20 |
| Low water level / High water level | | 2.4 | 3.7 | 2.43 |
| Low water level / Medium water level | | 2.7 | 2.3 | -2.0 |
| Wet | C1-H-Wet C2-H-Wet C5-M-Wet | 0.48 ± 0.83 | 1.46 ± 3.3 | 1.19 ± 48 |
| Dry | C3-L-Dry C6-M-Dry | 0.84 ± 0.98 | 2.93 ± 3.8 | 22.6 ± 108 |
| Dry-wet transition | C4-L-DWT | 0.94 ± 0.82 | 3.709 ± 3.9 | 21.5 ± 43 |
| Dry/wet | | 1.8 | 2.0 | 19 |
| Dry/Dry-wet transition | | 0.90 | 0.8 | 1.0 |

concentrations of ~ $5 \times 10^{-2}$ to $8 \times 10^{-1}$ μmol L$^{-1}$ measured during the rainy periods of May and Nov/2018 (C2-H-Wet and C5-M-Wet), as well as during the dry period in Feb/2019 (C6-M-Dry) (S3 Fig).

Generally, surface CH$_4$ concentrations ranged around $3 \times 10^{-1}$ μmol L$^{-1}$, being lower than those in the hypolimnetic over-plume layer OPL (Figs 5 and S3). During the rainy season (medium-level-wet campaign C5-M-Wet), concentrations were up to one order of magnitude lower, and no bottom CH$_4$ peaks were observed (S3 Fig). This suggests minimal vertical transport of CH$_4$ from the bottom to epilimnetic waters during this period.

Some exceptionally high CH$_4$ values, ranging from ~ $6 \times 10^1$ (at the dam zone, morning sample, P1-M) to ~ $4 \times 10^2$ μmol L-1 (at the inflow zone, P3), were observed at the surface. Two of these occurrences took place during the unusual final medium-level-dry campaign in Feb/2019 (C6-M-Dry) (Fig 4 –panel E4) under low wind speed ( < 1.8 m s$^{-1}$ according to meteorological records and zero according to field notes) (S4 Fig). This suggested a possible accumulation of dissolved CH$_4$ in the surface layer, which, due to low wind speed and consequently reduced surface turbulence, was not efficiently degassed to the atmosphere. Following the elevated values measured in the final campaign at the dam zone (morning sample, C6-M-Dry, P1-M), typical concentrations were observed during the afternoon sample (P1-A), right after a particularly intense rain event (S4F Fig). Even excluding these exceptional values, CH$_4$ concentrations in the surface were, on average, 2.4 times higher during low water levels compared to high and medium water levels (Table 3).

## Turbulent dissipation rates and diffusivity in the SML

The mean dissipation rates of turbulent kinetic energy at the water surface ($\epsilon_m$) ranged from ~$10^{-9}$ to ~$10^{-6}$ m$^2$ s$^{-3}$ and the estimated turbulent diffusivity ($K_z$) ranged between ~$10^{-5}$ and ~$10^{-4}$ m$^2$ s$^{-1}$ (S3 Table). The maximum measured $\epsilon_m$ of ~$10^{-6}$ m$^2$ s$^{-3}$ occurred during the low-level-dry-to-wet-transition campaign C4-L-DWT in the presence of strong wind speed ($U_{10}$ >

6 m s$^{-1}$) (S3 Table), showing the high sensitivity of near-surface turbulence to the dynamic atmospheric conditions, which makes it difficult to identify seasonal patterns in episodic measurements.

In the mid-lake (P2) and dam (P1) zones, both vertical eddy diffusivity ($K_z$) and dissipation rate of turbulent kinetic energy ($\epsilon_m$) were highest in the surface layer. Conversely, in the inflow zone (P3), both turbulence parameters were higher in the bottom-following plume than at the surface, except during the low-level-dry-wet transition campaign (C4-L-DWT), when $K_z$ and $\epsilon_m$ peaked at the surface due to strong wind (S3 Table, S5 Fig). This implies a relatively higher contribution of vertical transport of dissolved gases from the bottom to the surface in the inflow zone, and a relatively higher potential for gas evasion at P2 and P1.

Following persistent strong wind speed, high $\epsilon_m$ and $K_z$ from the surface extended down to the thermocline during afternoon samplings of the dry and dry-to-wet transition campaigns in Aug and Sept/2018 (C3-P1-A, C4-P1-A). Between the thermocline and 30 m depth, both $K_z$ and $\epsilon_m$ decreased in magnitude from P3 to P1 (S5 Fig), indicating the influence of the plume over the calm, thin upper layers in the inflow region (P3) and its diminishing impact towards the dam zone.

Diurnal variation of $\epsilon_m$ at the water surface were observed during all campaigns, with an increase of one to two orders of magnitude between morning and afternoon, when the wind speed increased, followed by a small decrease between the afternoon and the night. All minimum values of $\epsilon_m$ at the water surface in the dam zone (P1-M) were observed in the morning. The diurnal variability of $K_z$ was not as large in magnitude as the dissipation rates, but slightly higher values were observed in the afternoon measurements (S3 Table).

## Water-air gas fluxes and $k_{600}$

The diffusive fluxes of CH$_4$ ($F_{CH4}$) were always positive (evasion) and ranged from 0.046 to 2.3 mmol m$^{-2}$ d$^{-1}$, showing that the reservoir was a continuous source of CH$_4$ throughout the study period (S4 Table). The seasonal sampling revealed that $F_{CH4}$ was about four times higher during periods with low water level than during periods with high water level, when also surface concentrations ($C_{w\_CH4}$) were higher (Table 3). Moreover, $F_{CH4}$ was about twofold higher during the dry seasons than during wet periods. No consistent spatial pattern of the diffusive fluxes could be observed because most of the fluxes measured at the upstream section of the reservoir (inflow zone P3 and mid-lake zone P2) were rejected (S4 Table), suggesting interference of bubbles with the floating chamber measurements. Gas transfer velocities ($k_{600\_CH4}$) estimated from CH$_4$ fluxes ranged from 0.01 to 27.7 cm h$^{-1}$ (S4 Table) for wind speeds ($U_{10}$) between 0.5 and 7.8 m s$^{-1}$ (S4 Fig).

The diffusive fluxes of CO$_2$ varied from -53.0 to 320.6 mmol m$^{-2}$ d$^{-1}$. In daytime sampling (S4 Table), the fluxes were frequently negative, indicating that the reservoir functioned as a sink for atmospheric CO$_2$, with an average uptake of -25.1 mmol m$^{-2}$ d$^{-1}$ during periods of high photosynthetic activity. CO$_2$ uptake was more prevalent in the inflow and mid-lake zones of the reservoir (P3 and P2), where the proportion of negative fluxes was 77% and 100%, respectively, compared to 45% in the daylight samples at the dam zone P1. The timing of sampling significantly influenced the measured CO$_2$ fluxes, with late afternoon, evening, and early morning fluxes tending to be more positive (evasion). Conversely, during late morning, midday, and early afternoon, the fluxes tended to be lower and were frequently negative (uptake), suggesting a potential diurnal cycle of CO$_2$ fluxes (Fig 6).

Similar to CH$_4$ fluxes ($F_{CH4}$), also the fluxes of CO$_2$ ($F_{CO2}$) were higher during low-water-level and dry periods, reaching two-fold and 19-fold higher levels than during high-water levels and wet periods, respectively. However, the mean flux during the dry periods

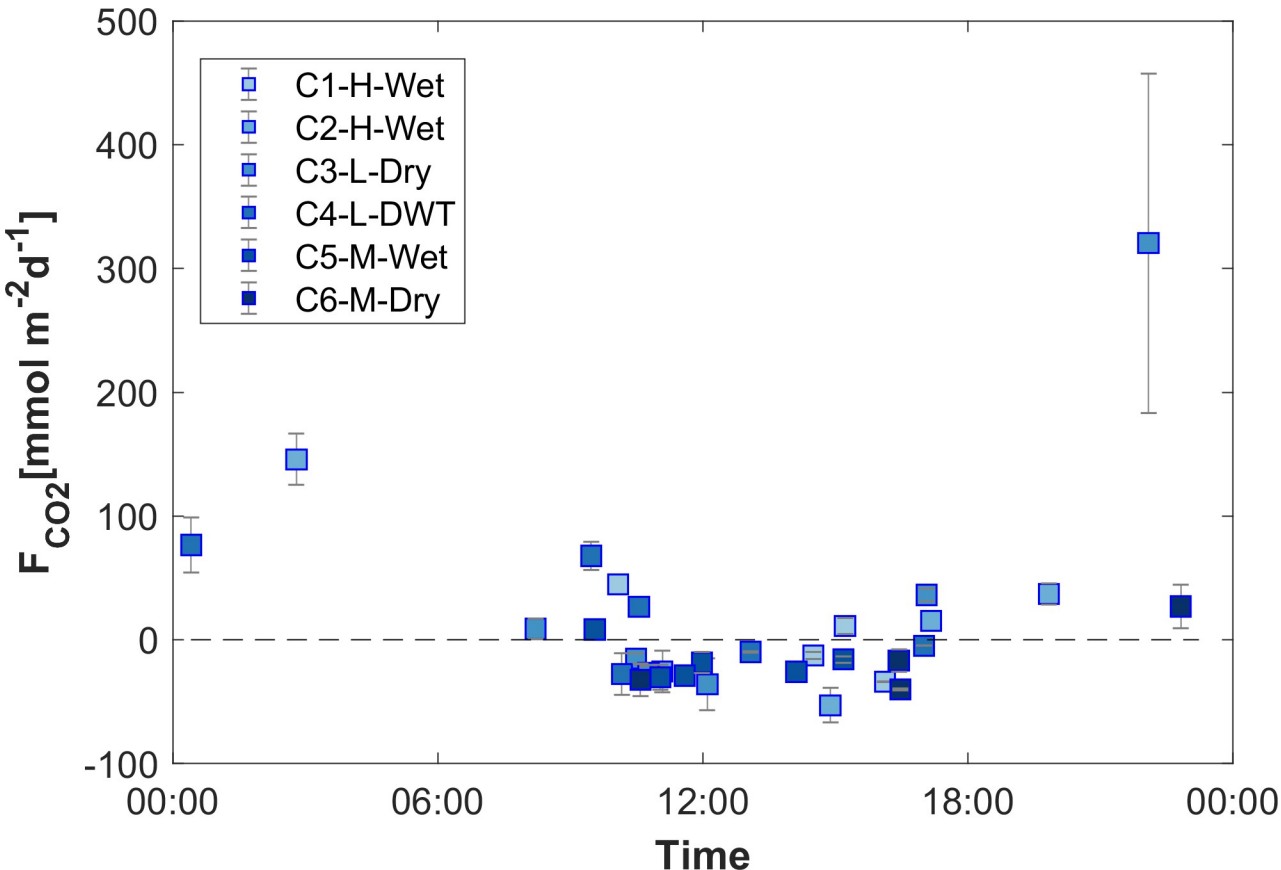

**Fig 6. Diurnal dynamics of CO$_2$ fluxes.** All accepted CO$_2$ flux estimates during the whole study period versus the time of day at which the measurements were conducted.

(22.6 ± 108 μmol m$^{-2}$ d$^{-1}$) was influenced by an exceptionally high flux recorded at the low-level-dry campaign in Aug/2018 (C3-L-Dry at 22:05) (321 mmol m-2 d$^{-1}$). Excluding this value, the average flux during dry periods was negative (-10.6 ± 28 μmol m$^{-2}$ d$^{-1}$), highlighting considerable variability in CO$_2$ fluxes during dry seasons and the significant impact of individual measurements on the mean flux estimate. The mean flux during the dry-wet transition period (21.5 ± 43) was similar to that of the dry periods (Table 3).

## CH$_4$ mass balances

The mass balances indicate that changes of surface water CH$_4$ concentration over time were mainly explained by *evasion* and *reaction* within the surface layer (Fig 7). During the wet seasons of May and Nov/2018, C2-H-Wet and C5-M-Wet, those terms were larger in the inflow and mid-lake zones of the reservoir (P3 and P2) (~10–6–10$^{-7}$ g s$^{-1}$) than in the dam zone (P1) (~10–7–10$^{-8}$ g s$^{-1}$) (Fig 7 panel A and D). During the dry and low-water-level campaigns (low-level-dry C3-L-Dry, low-level-dry-to-wet-transition C4-L-DWT and medium-level-dry C6-M-Dry) both terms were additionally high at daylight hours near the dam (Fig 7 panels B, C and E).

In the final campaign of Feb/2019, the medium-level-dry C6-M-Dry, the terms of the mass balance were exceptionally high (up to ~10$^{-4}$ g s$^{-1}$) at the inflow region P3 and the morning sample of the dam zone P1-M (Fig 7 panel E, S4F Fig). This may have been caused by a

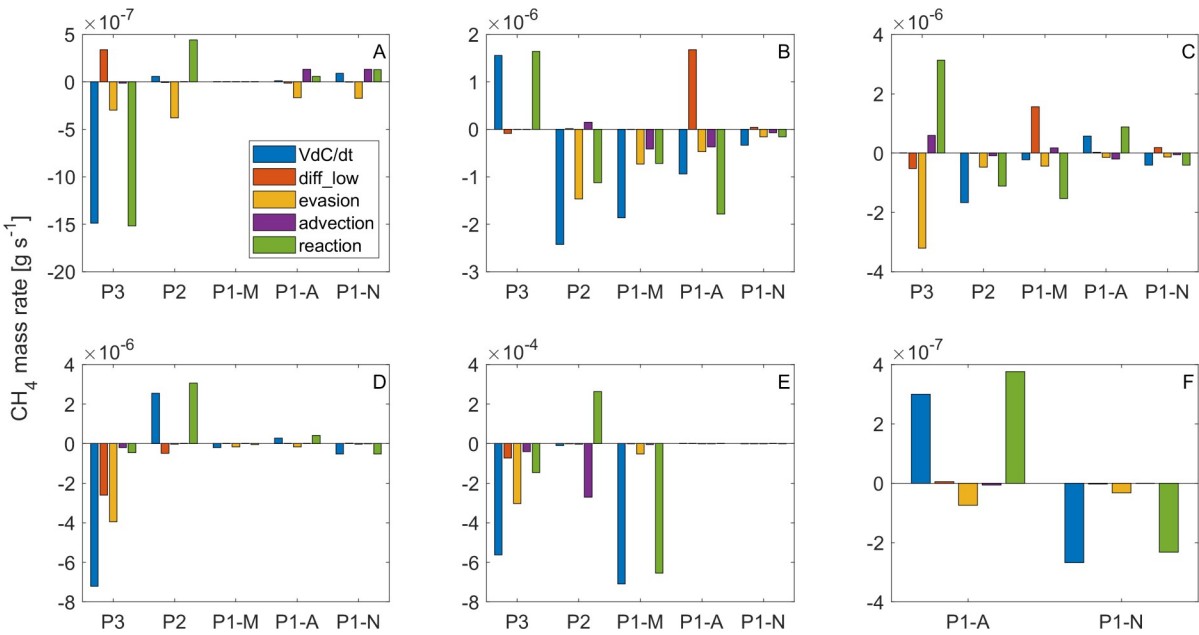

**Fig 7. Terms of the $CH_4$ mass balances at the surface across various sampling campaigns (panels).** A) campaign C2-H-Wet, B) campaign C3-L-Dry, C) campaign C4-L-DWT, D) campaign C5-M-Wet, E) campaign C6-M-Dry, F) shows a more detailed view on P1-A and P1-N of the final campaign C6-M-Dry. Scales of the subplots are different in order to observe all terms. Positive (negative) flux indicates towards (from) the water surface.

possible accumulation of $CH_4$ after a long low-wind period and to a large net methane production. In contrast, and regardless of the season, $CH_4$ mass balance components were always smaller at night (P1-N) (~$10-7-10^{-8}$ g s$^{-1}$) during all campaigns. Spatial and diel patterns were similar to those found for the trophic state indicators (pH and dissolved oxygen), suggesting that the hour-scale variations, *evasion* and *reactions* are favored by processes occurring during daylight and are stronger during periods with high photosynthetic activity.

There was significant transfer of dissolved $CH_4$ between the surface and lower layers (*diff_low*) when the thermocline exhibited strong turbulence and diffusivity and notable differences in $CH_4$ concentrations were detected between the surface and the hypolimnetic over-plume layer (OPL) (Fig 7 panels A to D, S3 Table). This pattern occurred in the inflow zone (P3) during the wet campaigns C2-H-Wet and C5-M-Wet, in the high-wind afternoon sampling at the dam zone (P1-A) during the low-level-dry campaign C3-L-Dry, and in the morning at the dam zone (P1-M) during the low-level-dry-to-wet campaign C4-L-DWT. The exchange involved the transport of $CH_4$ from the hypolimnion to the surface, driven by recurrently high $CH_4$ concentrations in the OPL (S3 Fig).

Particularly, the results obtained for the *evasion* term indicate that the inflow and mid-lake zones of the reservoir (P3 and P2) together are the most important sources of diffusive $CH_4$ fluxes to the atmosphere (Fig 7), which may be related to higher $CH_4$ concentrations at the water surface and also to the increasing turbulence intensity near the river inflow.

Reactions were always among the dominant terms of the balances and showed both net production and consumption of $CH_4$. The importance of the consumption/production processes did not show obvious seasonal or spatial patterns, but diel dynamics were observed in the lacustrine dam zone (P1). The recurrent negative sign of the reactions term suggests net $CH_4$ consumption (oxidation) in the morning and night (P1-M and P1-N) and net production in the afternoon (P1-A). Also, during the medium-level-dry campaign C6-M-Dry high

differences in concentrations between the mid-lake and the dam zones (P1 and P2) resulted in large advective fluxes and high reactions (Fig 7 panel E).

The Net Methane Production ($P_{net,CH4}$) varied from -3.0 µmol $L^{-1}$ $d^{-1}$ to 8.3 µmol $L^{-1}$ $d^{-1}$ across all campaigns, with the exception of the medium-level-dry C6-M-Dry. In this campaign, notably elevated magnitudes (ranging from -883 to 316 µmol $L^{-1}$ $d^{-1}$) were observed in the inflow zone (P3), mid-lake zone (P2), and the morning sample of the dam zone (P1-M) (S6 Fig), and were related to exceptionally high surface $CH_4$ concentrations measured in this campaign and discussed below. These local results reflect changes within a few hours resulting in very high estimates of consumption at P1-M. However, the increase in $CH_4$ concentrations from the other campaigns to the final campaign C6-M-Dry led us to believe that exceptional positive $P_{net,CH4}$ occurred during the last period. Since the wind speed was low during the sampling of the dam zone in the night and morning (P1-N and P1-M), we assume that the only important terms of the balance were the accumulation and the reactions, that yield a $P_{net,CH4}$ of 49 µmol $L^{-1}$ $d^{-1}$ for accumulating 62 µmol $L^{-1}$ in the SML at the dam zone P1 within the 30 h.

## Discussion

### Trophic State and $CO_2$ flux

Eutrophic conditions in the reservoir during the whole study period was indicated by the high phytoplankton abundance shown by visual observation of water color during the field campaigns, the recurrently observed negative $CO_2$ fluxes ($F_{CO2}$) during daytime, the supersaturation and large diel variations in dissolved oxygen (DO), the hypoxic to anoxic conditions in the hypoliminion (<10% of saturation), and the consistently low Secchi disk depth (SD < 2 m) at most of the sampling stations [27]. This agrees with the Toledo's trophic state index (TSI > 54, company information) [28] (S7 Fig). Eutrophic conditions are mostly caused by nutrient enrichment [29,30], which is consistent with the high nutrients loads expected from the polluted inflowing Porce river. Eutrophic characteristics during the study period were more pronounced during May–Aug/2018 and weaker during Sep–Nov/2018 suggesting intra-annual changes in the nutrient availability. In addition, eutrophication showed a spatial pattern, with Secchi depth being consistently lower and the trophic state index being generally higher at the transition and inflow zones, compared to the dam zone. Highest concentration of chlorophyll-a around the inflow region have been observed in other tropical and subtropical reservoirs and were related this to the effect of wind pattern [14] and nutrient input from the river [31]. Following the same spatial pattern as the trophic condition, $CO_2$ uptake at Porce III was more frequently observed at the upstream section (inflow and transition zones) compared to the lacustrine dam region (S4 Table). The findings of [15] suggest that there is an important link between surface $CO_2$ and aquatic metabolism in tropical reservoirs, varying from under to supersaturated and alternating periods of $CO_2$ uptake and emission. A different spatial pattern emerged in an oligotrophic tropical reservoir, where consistent evasion (rather than uptake) of $CO_2$ from the water was observed, which gradually decreased from the inflow area to the lacustrine region [18]. Our results suggest that the spatially variability of $CO_2$ fluxes within the studied reservoir were influenced by the aquatic autotrophic metabolism.

Our findings indicate that $CO_2$ fluxes tend to be higher during dry seasons and when the water level was low. On average, the $CO_2$ flux was 2.4-fold higher at low water levels compared to high water levels and 19-fold higher in dry than in rainy seasons. However, it is important to note that there is significant variability in $CO_2$ flux measurements, making them highly sensitive to individual data points that exhibit exceptionally high flux values. The decrease in water levels may be associated with reduced water retention time in the reservoir, which has been found to lead to higher and consistently positive $CO_2$ fluxes in another tropical reservoir

[14]. Similarly, increased fluxes during dry seasons driven by higher CO$_2$ surface concentrations were reported for tropical reservoirs and are influenced by factors such as depth and residence time [22,32]. This suggests that dam operation and hydrological conditions may have a significant impact on CO$_2$ emissions from tropical reservoirs.

Most published studies estimating CO$_2$ emissions from reservoirs have limited temporal resolution, often spanning from weeks to seasons, thus overlooking the daily variations in CO$_2$ fluxes [33]. The observation of alternating positive and negative CO$_2$ fluxes in the Porce III reservoir may indicate that the reservoir is at times both a sink and a source of CO$_2$ to the atmosphere. In fact, CO$_2$ fluxes observed at various seasons and times suggested a diurnal cycle, wherein CO$_2$ is released during periods of reduced solar radiation (evening and early morning) and absorbed during daylight hours when solar radiation is high (late morning, midday and early afternoon). This agrees with observations from subtropical aquatic systems where eutrophic waters could alternate between CO$_2$ sources and sinks during the span of a day [34,35]. More moderate behavior has been observed in other studies that have found significantly increasing CO$_2$ fluxes from daytime to nighttime in eutrophic subtropical reservoirs, but still consistent emissions [36]. Although very few observations are available for tropical reservoirs, diurnal dynamics in CO$_2$ fluxes from inland waters have been related to variations in surface heat exchange between day and night [34,37] and in biochemical processes, such as the balance between photosynthesis and respiration in the surface water, which are regulated by temperature and sunlight availability [38]. Thermal convection has the potential to amplify fluxes during cooling, potentially influencing the diurnal CO$_2$ cycle. However, diurnal pattern in gas transfer velocity would not alter the flux direction. The only circumstances under which the direction of air-water flux changes is when surface concentrations become oversaturated (resulting in a positive flux) or undersaturated (leading to a negative flux) relative to the atmospheric equilibrium concentration. Therefore, we infer that autotrophic metabolism is likely to be the main driver for the diurnal cycle of the CO$_2$ flux, as concluded in previous studies in subtropical eutrophic aquatic systems [35,39].

## Surface CH$_4$ dynamics

The diffusive fluxes of CH$_4$ ($F_{CH4}$, 0.046–2.3 mmol m$^{-2}$ d$^{-1}$) were in the range of fluxes reported for other tropical reservoirs [15,24,40]. Although some of our observations exceeded this range, the utilization of the mechanistic surface renewal model indicated that these values were physically not plausible and the measurements possibly affected by gas bubbles (SI "Bubble detection"). Therefore, we only considered data points where $k_{600}$ was less than or equal to 30 cm h$^{-1}$, while values exceeding this threshold were attributed to gas bubbling into the chamber.

We found that during low water level, the CH$_4$ fluxes were up to 3.7 times higher compared to high water level seasons. Only a limited number of studies have investigated the seasonal fluctuations of diffusive methane emissions from tropical reservoirs. It is not meaningful to compare the current findings with high-latitude systems, as their hydrological patterns differ significantly. Tropical systems experience distinct rainy and dry seasons, and the reservoirs maintain a weak stratification throughout the year. In contrast, temperate systems typically exhibit strong stratification in summer and complete mixing in autumn and winter, while boreal systems are iced-covered during winter. In agreement with our results, higher CH$_4$ emissions during dry seasons were also reported by [24] in a tropical reservoir. Dry seasons are normally associated with low water levels, resulting in faster transport of CH$_4$ produced in the sediment to the water surface and reduced CH$_4$ oxidation. On the other hand, some authors have not found seasonal patterns of CH$_4$ fluxes in other tropical reservoirs [22].

Excluding the extreme values ($> 10^1$ µmol L$^{-1}$), the measured CH$_4$ surface concentrations ($C_{w\_CH4}$) were comparable to values previously reported for the epilimnion in other tropical reservoirs [15,24,41,42]. The upper limit (3 µmol L$^{-1}$) is at the high end of concentrations reported for stratified waters, even if compared to values reported for tropical reservoirs with high organic load, such as Furnas Reservoir in Brazil [15]. However, values of similar magnitude, even as high as our extreme values, have been reported in other systems close to the epilimnion and have been attributed to oxic methane production (OMP) [43].

The lowest dissolved CH$_4$ concentration in the surface was found during the medium-level-wet season campaign (C5-M-Wet), when the river inflow was along the bottom of the reservoir and no anoxic bottom layer was present. No extreme peaks of CH$_4$ at larger depths ($> 4 \times 10^1$ µmol L$^{-1}$, see profiles in S3 Fig) were observed under these conditions. This may be associated with CH$_4$ oxidation [17,44] during the vertical transport of the gas from the sediment to the surface in the fully oxic water column during this campaign, as opposed to the other campaigns, in which a thick anoxic layer at the bottom might have resulted in less CH$_4$ oxidation.

The concentration of dissolved methane at the water surface ($C_{w\_CH4}$) was relatively constant across the reservoir during periods of high water levels. However, there was a consistent decrease in concentration from the inflow and mid-lake zones towards the dam zone during low water levels. The elevated concentrations of CH$_4$ in the Porce III reservoir during specific seasons were linked to hydrological conditions prevalent during low water levels. This aspect of CH$_4$ dynamics in tropical reservoirs is relatively unexplored, as seasonal dynamics have received limited attention in previous research, which primarily focused on climatic conditions rather than variations in reservoir water levels. In line with this, higher CH$_4$ concentrations in the epilimnion during dry seasons compared to wet seasons were observed in Petit Saut reservoir [24].

The increased concentrations of CH$_4$ in the inflow region and mid-lake zones of the reservoir aligned with the more pronounced eutrophic conditions in these zones. This condition was characterized by higher levels of dissolved oxygen and pH, along with a lower Secchi depth compared to the dam zone. The surface CH$_4$ pool is influenced by various carbon sources and processes, including metabolism, riverine inputs, and sediment contributions [18]. Consequently, the breakdown of sedimented phytoplankton can promote CH$_4$ production in strongly eutrophic zones [45]. Additionally, the observed spatial pattern in the reservoir may be linked to the decomposition of allochthonous carbon deposited in the inflow region, where delta formation is common and sediment deposition favors anaerobic metabolism [46,47].

There is a traditional association of CH$_4$ production to anaerobic processes. However, besides the spatial pattern of higher methane concentrations towards the more oxic zone, the mass balances revealed frequent occurrence of net CH$_4$ production in the surface layer (positive reaction rates). Several studies have shown that the trophic state is positively correlated to CH$_4$ emissions and/or concentrations in reservoirs [1,15,48]. This is traditionally associated with low oxygen concentration and high content of autochthonous carbon in the hypolimnion, both promoting CH$_4$ production [49]. Recent research [18] has revealed the possible production of CH$_4$ in the oxic euphotic zone of lakes and reservoirs, recurrently showing peaks in the epilimnion or near the water surface [50–53]. Oxic methane production (OMP) has been associated with the autotrophic activity since it has been positively correlated to photosynthesis, oxygen concentrations and dissolved organic carbon [50,54] suggesting a link between OMP and algal derived substrates available for methanogens [50,52,53] and even, the direct CH$_4$ production by phytoplankton as suggested by [55]. Other explanations point toward non-microbial CH$_4$ production by photochemical reactions [56,57]. However, OMP rates reported for laboratory incubations and natural oligo-mesotrophic lakes (50 to ~200 nmol L$^{-1}$ d$^{-1}$) [53] are about one to five orders of magnitude lower than our estimates.

Negative values of reaction rates (indicating consumption in the SML) obtained from the mass balances are expected to result from microbial CH$_4$ oxidation (MOx), which is occurring under oxic conditions in the epilimnion and hypolimnion [17]. Excluding the very particular medium-level-dry campaign C6-M-Dry in Feb/2019, our net consumption rates are 2 or 3 orders of magnitude higher than MOx rates, reported for natural oligo-mesotrophic lakes (4–60 nmol L$^{-1}$ d$^{-1}$) [50,53] and oligotrophic tropical reservoirs (26 nmol L$^{-1}$ d$^{-1}$) [18], while the exceptional high consumption rate during the campaign C6-M-Dry is in agreement with oxidation rates found by [17] in the water column of a tropical eutrophic reservoir (134–1600 μmol L$^{-1}$ d$^{-1}$).

Clearly, we found a very wide range and little consistency in the prevalence of both competing processes (MOx and OMP), keeping open the discussion to the complexity of interactions between drivers/inhibitors of CH$_4$ metabolism under eutrophic conditions and extreme periods such as the campaign of Feb/2019 C6-M-Dry when the dry season conditions were exacerbated by El Niño phenomena and unusual short-term water level variations variations (Fig 2). For example, the estimated methane production near the dam during the final campaign in Feb/2019 (~50 μmol L-1 d$^{-1}$) are contradicted by a methane loss at the dam zone when the concentration decreased between the morning (P1-M) and the afternoon (P1-A) sampling (Figs 7 and S4F). The second measurement, however, was made after a very intense rain event (S4F Fig) where $k_{600}$ may have increased significantly [58] and dissolved methane may have been evaded rather than biochemically consumed. Although fluxes were not measured during the rain event, it seems likely that short-term events such as rainfall may play an important role in greenhouse gas emissions and budgets in lakes and reservoirs, as other authors have found in field and experimental studies [24,58,59]. There is also scientific evidence that other highly variable factors may be involved in CH$_4$ dynamics such as oxygen concentrations and light exposure that have been shown to inhibit MOx [44,52].

## Conclusions

Measurements in a eutrophic tropical reservoir indirectly indicate that the surface layer is a region of intense microbial activity, where alternating oxic production and oxidation likely play a significant role in the CH$_4$ budget. Autotrophic activity also strongly influences the dynamics of CO$_2$ fluxes in this zone. The surface dynamics of greenhouse gases (GHG) generally undergo seasonal changes, and the various components contributing to the surface methane budget are influenced by hydrological and meteorological conditions. This suggests that understanding the surface processes involved in CH$_4$ dynamics in eutrophic tropical systems should be evaluated on a seasonal scale. Specifically, the water level appears to be crucial, as observed in Porce III reservoir, where CH$_4$ accumulation in the surface layer was higher at lower water level. Additionally, both CO$_2$ and CH$_4$ concentrations and fluxes showed daily fluctuations, emphasizing the importance of considering these diurnal dynamics in GHG emission estimates, as sampling only during the daytime could introduce bias to the results.

Spatially, the magnitudes of CH$_4$ reactions, surface concentrations, and emissions where highest in the inflow and the mid-lake zones of the reservoir, where also the eutrophic condition were more prevalent. This is consistent with recent studies associating photosynthetic activity with methane production. Our data suggested the frequent occurrence of oxic methane production (OMP) in the surface layer, especially in extremely dry periods. This highlights the complexity of interactions between drivers/inhibitors of CH$_4$ metabolism under eutrophic conditions. Future studies should reduce the uncertainty in CH$_4$ budgets by direct measurements of the net production rates of CH$_4$ in the surface mixed layer.

## Supporting information

**S1 Appendix.**
(PDF)

**S1 Fig. 30-day-moving average of selected meteorological variables and surface water temperature during the study period (1 h resolution). a**. Water surface (blue) and air (red) temperature, **b**. wind speed, **c**. solar radiation and **d**. cloud cover fraction. Grey vertical bars mark the time of the field campaigns.
(PDF)

**S2 Fig. Physicochemical profiles and vertical structure of Porce III reservoir.**
(PDF)

**S3 Fig. Dissolved $CO_2$ and $CH_4$ profiles during the high-level-wet campaign C1-H-Wet.** The background color represents the layers according to the previously defined conventions (Fig 1B–main manuscript). Hidden values: right panel 399 μmol $L^{-1}$ at 115 m depth.
(PDF)

**S4 Fig. Surface concentration of $CH_4$ and diffusive fluxes and several forcings.**
(PDF)

**S5 Fig. $\epsilon_m$ and $K_z$ profiles.**
(PDF)

**S6 Fig. Estimated net methane production ($P_{net,CH4}$) in the surface at different sampling locations and times (P1, P2 and P3) and for the campaigns. a**. high-level-wet C2-H-Wet, b. low-level-dry C3-L-Dry, c. low-level-dry-to-wet-transition C4-L-DWT, d. medium-level-wet C5-M-Wet e. medium-level-dry C6-M-Dry, f. shows a more detailed view on P1-A and P1-N of the final campaign C6-M-Dry. Scales of the subplots are different in order to observe all terms. Positive (negative) bars indicate production (consumption) of CH4 at the SML.
(PDF)

**S7 Fig. Trophic state index (TSI) See locations of sampling stations E1 to E4 in Fig 1A (main manuscript).** Source: Empresas Públicas de Medellín–EPM.
(PDF)

**S1 Table. Porce river inflow characteristics during the study period.** Surface temperature (T), electrical conductivity (EC), dissolved oxygen (DO), pH, Ammonia ($NH_3^-$), nitrites $NO_2^-$, nitrates ($NO_3^-$), phosphates $PO_4^3$ and sulfates ($SO_4^-$).
(PDF)

**S2 Table. General description of the field campaigns.** The ID campaign combines a campaign number (C1 to C6), water level condition (high level "H", low level "L", medium level "M") and the seasonal hydrological condition (Wet season "Wet", Dry-wet transition "DWT", Dry season "Dry").
(PDF)

**S3 Table. Mean dissipation rates of turbulent kinetic energy ($\epsilon$) and turbulent diffusivity ($K_z$) in the layers: surface mixed layer (SML), Thermocline (Therm.), Over-plume layer (OPL), river plume (Plume) and Bottom.**
(PDF)

**S4 Table. Atmospheric fluxes and surface concentrations of $CO_2$ and $CH_4$ observed during chamber measurements and estimated gas transfer velocities.** Water surface temperature

during chamber deployments (T), atmospheric fluxes of CO$_2$ ($F_{CO2}$), atmospheric fluxes of CH$_4$ ($F_{CH4}$), water surface concentration of CO$_2$ ($C_{w\_CO2}$), water surface concentration of CH$_4$ ($C_{w\_CH4}$) and the gas transfer velocity estimated from chambers ($k_{600\_CH4}$). N.A. are not accepted data after quality control.
(PDF)

## Acknowledgments

Thanks to Empresas Públicas de Medellín (EPM) for its invaluable support during the field work and supporting information. We are grateful to Victoria Ramírez-Rivera, César Augusto Jaramillo-Gutiérrez, Lina María Ramírez-Morales, Ricardo Román-Botero, Óscar Darío Beltrán-Pérez and Andrés Felipe Monsalve-Salazar who participated in the field work and laboratory analysis. Thanks also to Rockland Scientific Inc. for the constant support microCTD data processing.

## Author Contributions

**Conceptualization:** Eliana Bohórquez-Bedoya, Juan Gabriel León-Hernández, Andreas Lorke, Andrés Gómez-Giraldo.

**Data curation:** Eliana Bohórquez-Bedoya.

**Formal analysis:** Eliana Bohórquez-Bedoya, Juan Gabriel León-Hernández, Andrés Gómez-Giraldo.

**Funding acquisition:** Juan Gabriel León-Hernández, Andrés Gómez-Giraldo.

**Investigation:** Eliana Bohórquez-Bedoya, Juan Gabriel León-Hernández, Andrés Gómez-Giraldo.

**Methodology:** Eliana Bohórquez-Bedoya, Juan Gabriel León-Hernández, Andreas Lorke, Andrés Gómez-Giraldo.

**Project administration:** Andrés Gómez-Giraldo.

**Supervision:** Juan Gabriel León-Hernández, Andreas Lorke, Andrés Gómez-Giraldo.

**Validation:** Eliana Bohórquez-Bedoya.

**Visualization:** Eliana Bohórquez-Bedoya.

**Writing – original draft:** Eliana Bohórquez-Bedoya.

**Writing – review & editing:** Andreas Lorke, Andrés Gómez-Giraldo.

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
