## [Decision Letter · Decision Letter 0]

12 Sep 2023

PONE-D-23-24316Surface CO2 and CH4 dynamics in a eutrophic tropical Andean reservoirPLOS ONE

Dear Dr. Lorke,

Thank you for submitting your manuscript to PLOS ONE. After careful consideration, we feel that, while it has merit, it does not meet PLOS ONE’s publication criteria. Therefore, we invite you to submit a completely revised version of the manuscript that addresses all the points raised during the review process.

We look forward to receiving your revised manuscript.

Kind regards,

Steven Arthur Loiselle

Academic Editor

PLOS ONE

Journal Requirements:

3. We noted in your submission details that a portion of your manuscript may have been presented or published elsewhere. "The essential content of this article is an integral part of the doctoral dissertation “Physical processes influence on the dynamics of the main greenhouse gases in mountain tropical reservoirs” in co-tutelle agreement submitted to the Universidad Nacional de Colombia (Colombia) and the University of Kaiserslautern-Landau (Germany). " Please clarify whether this conference proceeding or publication was peer-reviewed and formally published. If this work was previously peer-reviewed and published, in the cover letter please provide the reason that this work does not constitute dual publication and should be included in the current manuscript.

4. We note that Figure 1 in your submission contain map/satellite images which may be copyrighted. All PLOS content is published under the Creative Commons Attribution License (CC BY 4.0), which means that the manuscript, images, and Supporting Information files will be freely available online, and any third party is permitted to access, download, copy, distribute, and use these materials in any way, even commercially, with proper attribution. For these reasons, we cannot publish previously copyrighted maps or satellite images created using proprietary data, such as Google software (Google Maps, Street View, and Earth). For more information, see our copyright guidelines: http://journals.plos.org/plosone/s/licenses-and-copyright.

Additional Editor Comments: 

The study and resulting manuscript are important, however, there are major shortcomings in the objectives (hypothesis), methods and results. The analysis is not sufficiently complete and the overall organisation of the manuscript does not meet the requisites of the journal. If the authors decide to resubmit the manuscript, it will need to be completely revised, addressing all the shortcomings identified by the reviewers.

Reviewers' comments:

Reviewer's Responses to Questions

**Comments to the Author**

1. Is the manuscript technically sound, and do the data support the conclusions?

Reviewer #1: Yes

Reviewer #2: Partly

2. Has the statistical analysis been performed appropriately and rigorously? 

Reviewer #1: Yes

Reviewer #2: N/A

3. Have the authors made all data underlying the findings in their manuscript fully available?

Reviewer #1: Yes

Reviewer #2: Yes

4. Is the manuscript presented in an intelligible fashion and written in standard English?

Reviewer #1: Yes

Reviewer #2: Yes

5. Review Comments to the Author

Reviewer #1: The work brings an important nandynamic contribution of eutrophic methane emissions from Tropical Andean Reservoir using a type of mass transfer model and experimental measurements to predict the behavior of methane emissions.

The introduction brings a bibliographic reference to justify the research problem, but the hypothesis that the researchers tested is not clear, that is, what would be expected for this environment and for these spatial and seasonal conditions.

I also think that a more detailed characterization of the study area is needed. Why is this region eutrophic? what aspects of occupation do have more nutrients?

a more detailed comparison with other works published in tropical areas is advisable.

Reviewer #2: Manuscript entitled Surface CO2 and CH4 dynamics in a eutrophic tropical Andean reservoir by Bohórquez-Bedoya et al. report results of six measurement campaigns of CO2 and CH4 fluxes and water concentrations and physicochemical and meteorological parameters in three locations along a narrow canyon reservoir in Peru. Some campaigns also included diurnal sampling. The manuscript has potential as when aiming to answer questions which factors drive lateral and vertical dynamics of greenhouse gases CO2 and CH4 in a reservoir. Physical forcings affecting the lateral and vertical gas transfer was a key part of the manuscript. Authors calculated mass balance for CH4 and estimated microbial processes indirectly. River-like reservoirs are common globally, but according to authors, GHG dynamics are not studied in such reservoirs in the tropical zone. Both spatial and temporal coverage were scarce, though diurnal dynamics was included in the campaigns. Temporally the sampling may be okay in tropical conditions where seasonality is less pronounced than in higher latitudes. There were, however, some shortcomings. It was mentioned that ebullition was captured in some chamber measurements of CH4, but rejected from further analyses and thus ebullition was otherwise omitted. Partly the methods were poorly described.

In its current form, to my opinion, the manuscript is premature to be considered for publication. It didn’t help that the text was hard to read. Personally, I found it hard to keep track of the campaigns and locations’ abbreviations and which data were considered for which purposes. I liked the graph in the supplement showing the sampling moments in relation to hydrological cycles. In all, graphs in the supplement had better quality. Authors could rewrite the text considering clarity. I think there is space for further digestion of the data because the main document contained 13 figures and the supplement had an additional 23 figures. At a quick glance, some figures in the supplement could be relevant/better in the main document but I recommend authors to re-think how to present the data with reasonable number of figures and tables (<10). Reviewing would have been much smoother if the figures had been in an order and accompanied the legend text. Below are some detailed comments.

L 22. Why surface in the title and here? Concentrations and physicochemical parameters were studied across the water column.

L 63. dissolved what?

L. 77. or thereabouts. Could you formulate hypotheses?

L. 111. water quality is related to the hydrological status?

L. 120. Clarify the diurnal sampling. Where, how frequently etc.

L. 153. How stable the water layers were?

L. 176. Headspace technique …and concentrations were analyzed using a GC

L. 228. How long is the chamber closure? In total 5 samples per closure?

L. 232. How were the vials prepared? Vacuumed? Flushed with N2? Was the amount sufficient for the GC?

L 241. Why this threshold value?

L. 511. could you show CH4 and CO2 fluxes in the same figure, with O2 and pH?

L. 517. think how to average fluxes in the cases that diurnal dynamics. What’s the proportional effect of diurnal dynamics.

L. 530. velocities based on measurements.

L. 532. correlation with estimates based on measurements. Be precise, and explain what is acceptable correlation?

L. 616. balance was calculated. Estimated?

L. 633. estimated net CH4 production...

L. 640. how was it observed? I suggest editing.

L. 687. if not corrected with alkalinity and pH value?

L. 692. basic > alkaline?

L. 727. the shallower water body the higher CH4 fluxes , a well reported and typical pattern in lakes

L 818. indirectly demonstrate? Did you measure microbial activity?

L- 820 Diurnal only > what you mean? Sampling over a 24h cycle only versus over a year?

6. PLOS authors have the option to publish the peer review history of their article (what does this mean?). If published, this will include your full peer review and any attached files.

Reviewer #1: No

Reviewer #2: No

---

## [Author Response · Author response to Decision Letter 0]

6 Dec 2023

Response to comments on the revised manuscript

Response to editors

If applicable, we recommend that you deposit your laboratory protocols in protocols.io to enhance the reproducibility of your results. Protocols.io assigns your protocol its own identifier (DOI) so that it can be cited independently in the future. For instructions see: https://journals.plos.org/plosone/s/submission-guidelines#loc-laboratory-protocols. 

Additionally, PLOS ONE offers an option for publishing peer-reviewed Lab Protocol articles, which describe protocols hosted onprotocols.io. Read more information on sharing protocols at https://plos.org/protocols?utm_medium=editorial-email&utm_source=authorletters&utm_campaign=protocols

The coauthors consider that publishing laboratory protocols is unnecessary in this case, as we employed generic protocols for analysis that have already been documented by other researchers.

Please ensure that your manuscript meets PLOS ONE's style requirements, including those for file naming. The PLOS ONE style templates can be found at https://journals.plos.org/plosone/s/file?id=wjVg/PLOSOne_formatting_sample_main_body.pdf and https://journals.plos.org/plosone/s/file?id=ba62/PLOSOne_formatting_sample_title_authors_affiliations.pdf

This submission meets Plos One’s style requirements based on the templates and on-line guides.

We noted in your submission details that a portion of your manuscript may have been presented or published elsewhere. "The essential content of this article is an integral part of the doctoral dissertation “Physical processes influence on the dynamics of the main greenhouse gases in mountain tropical reservoirs” in co-tutelle agreement submitted to the Universidad Nacional de Colombia (Colombia) and the University of Kaiserslautern-Landau (Germany). "Please clarify whether this conference proceeding or publication was peer-reviewed and formally published. If this work was previously peer-reviewed and published, in the cover letter please provide the reason that this work does not constitute dual publication and should be included in the current manuscript.

Certainly, the inclusion of the core material from the submitted article in a doctoral thesis signifies its availability in university archives, specifically those of Universidad Nacional de Colombia campus Medellín and the University of Kaiserslautern – Landau in Germany. However, it's important to note that such document serve as an academic prerequisite and do not qualify as peer-reviewed work.

We note that Figure 1 in your submission contain map/satellite images which may be copyrighted. All PLOS content is published under the Creative Commons Attribution License (CC BY 4.0), which means that the manuscript, images, and Supporting Information files will be freely available online, and any third party is permitted to access, download, copy, distribute, and use these materials in any way, even commercially, with proper attribution. For these reasons, we cannot publish previously copyrighted maps or satellite images created using proprietary data, such as Google software (GoogleMaps, Street View, and Earth). For more information, see our copyright guidelines: http://journals.plos.org/plosone/s/licenses-and-copyright.

Figure 1 was replaced by one that complies with the CC BY 4.0 license.

Response to Reviewer #1 

The introduction brings a bibliographic reference to justify the research problem, but the hypothesis that the researchers tested is not clear, that is, what would be expected for this environment and for these spatial and seasonal conditions. 

We add the following sentence at the end of the introduction section: “We hypothesized that the dynamics of greenhouse gases within a eutrophic reservoir are subject to seasonal variations in response to changes in hydrological conditions, and that the spatial distribution of these gases varies with proximity to the river inflow” (L 69-72).

I also think that a more detailed characterization of the study area is needed. Why is this region eutrophic? what aspects of occupation do have more nutrients?

In the initial version of the manuscript, the reservoir's eutrophic condition received inadequate attention in the study site section and was scattered throughout the document, with significant mentions in the introduction, methods, and discussion sections. In the revised version, we have incorporated details explaining the eutrophic nature of the region, relocated a table containing crucial study zone aspects (such as air and water temperatures) from the results section to the study site section, and also moved a sentence related to the trophic state index from the discussion section to the study site section:

“The river is dammed upstream in the Porce II reservoir and reaches Porce III with significant concentrations of nitrate, ammonium and phosphate (S1 Table). These inflow characteristics promote eutrophic conditions in both Porce II and Porce III reservoirs, as does the consistently high atmospheric and water temperature of around 25°C (Table 1), which favors algae growth. The trophic state index (TSI) (Toledo et al., 1983) estimated from the monitoring conducted by the reservoir operator, indicates constantly eutrophic condition of Porce III reservoir between 2016 and 2018 (TSI >54) (S22 Fig)”. (L91-98)

Concerning the second question, which aspects of human activity contribute more nutrients, it is unfortunate that we lack a comprehensive characterization of the diverse pollution sources within the study region. This lack of data hinders our ability to discern the relative contributions of different sources of nutrient input into the reservoir. Nevertheless, we do note that the primary river inflow into the reservoir receives wastewater from a large urban area, as well as effluents from industrial and agricultural activities.

A more detailed comparison with other works published in tropical areas is advisable.

In fact, information on greenhouse gas dynamics in tropical reservoirs is less abundant than in other latitudes. Moreover, several studies carried out in tropical reservoirs seek to quantify emissions mainly from statistical approaches, with only few mechanistic approaches. Additionally, there is a great scarcity of studies in tropical systems that analyze the temporal variability of GHGs, especially methane and there is no information (to our knowledge) available from tropical reservoirs regarding the diurnal cycle of CO2 surface fluxes. However, following the reviewer’s suggestion, we expanded this discussion. Indeed we found recent publications (from the last few years) that actually allowed us to make an interesting discussion of our results, some of them, from subtropical reservoirs (Lin et al., 2019; Paranaíba et al., 2021; Roland et al., 2010; Yang et al., 2022).

Response to Reviewer #2 

Manuscript entitled Surface CO2 and CH4 dynamics in a eutrophic tropical Andean reservoir by Bohórquez-Bedoya et al. report results of six measurement campaigns of CO2 and CH4 fluxes and water concentrations and physico chemical and meteorological parameters in three locations along a narrow canyon reservoir in Peru. Some campaigns also included diurnal sampling. The manuscript has potential as when aiming to answer questions which factors drive lateral and vertical dynamics of greenhouse gases CO2 and CH4 in a reservoir. Physical forcings affecting the lateral and vertical gas transfer was a key part of the manuscript. Authors calculated mass balance for CH4 and estimated microbial processes indirectly. River-like reservoirs are common globally, but according to authors, GHG dynamics are not studied in such reservoirs in the tropical zone. Both spatial and temporal coverage were scarce, though diurnal dynamics was included in the campaigns. Temporally the sampling may be okay in tropical conditions where seasonality is less pronounced than in higher latitudes. There were, however, some shortcomings. 

As mentioned by the reviewer, our study aimed to assess the various factors influencing greenhouse gas emissions in the studied reservoir, typically associated solely with biochemical aspects. In our investigation, we acknowledge physical forcing as a precursor for horizontal and vertical transport of dissolved gases.

In terms of sampling coverage, sampling coverage on the seasonal scale is considered sufficient for a tropical reservoir, given its lower seasonal variability compared to high-latitude systems (IHA, 2010). Considering the straightforward morphology of the studied reservoir—an elongated canyon reservoir—coupled with its modest size (4 km2), we propose three significant zones within the reservoir. These zones would be represented by three representative sampling points, enabling us to conduct more comprehensive measurements at each location. The latter included vertical profiling of physicochemical variables and turbulence measurements throughout the water column. Opting for three sampling points, as opposed to more, allowed us to conduct detailed monitoring at each location, minimizing the need for rapid assessments and reducing variability associated with temporal fluctuations.

It was mentioned that ebullition was captured in some chamber measurements of CH4, but rejected from further analyses and thus ebullition was otherwise omitted. 

The study aimed to examine diffusive fluxes throughout the reservoir, exploring the pathways through which dissolved gases reach the water surface or undergo production/transformation within it. However, a significant number of our measurements, conducted using floating chambers designed for diffusive fluxes, were influenced by bubbles. To detect these bubbles, we employed a methodology involving the estimation of the maximum physically conceivable flux based on a given concentration. Nevertheless, due to the pronounced variability inherent in ebullition fluxes, the sample size falls short of facilitating a thorough analysis of the phenomenon. Therefore, bubbling has not been widely addressed in the main manuscript, but an explanation of the methodology for bubbles detection is in the Supplementary information “Bubbles detection”. 

Partly the methods were poorly described.

Initially, our manuscript extensively detailed the physical processes related to the transport of greenhouse gases (GHG) in reservoirs, but the description of GHG sampling and laboratory analysis was lacking. In the revised version, we have addressed this by giving equal significance to both the field methods and the procedures involved in sampling, processing, and post-processing of data. For this, we have added details to the sections “sampling and data overview” (L. 117 -122, 140-146), “dissolved CO2 and CH4 in the water column” (L. 182-194), “air-water fluxes of CO2 and CH4 and k600” (L. 223-227) and have shorten the sections “meteorological and hydrological data collection”.

In its current form, to my opinion, the manuscript is premature to be considered for publication. It didn’t help that the text was hard to read. Personally, I found it hard to keep track of the campaigns and locations’ abbreviations and which data were considered for which purposes

Determining the optimal approach for presenting results and discussions was challenging due to the complexity of having three sampling points, diurnal and seasonal dynamics, and spatial variability (horizontal and vertical layers). Initially, we adhered to conventions established at the manuscript's methods. However, the reviewer has indicated that even with this approach, the text remains difficult to follow. In response to this feedback, we have extensively revised the manuscript to provide detailed descriptions of the sampling locations and campaigns, aiming for a more comprehensive understanding. For instance, in the initial version, we stated "...x condition was observed in P1-M of C3-L-Dry," whereas in the revised version, we now express it as "...x condition was observed during the low-level-dry campaign C3-L-Dry in the morning sample of the dam zone P1". Also, to the vertical layering we now refer in full words.

I liked the graph in the supplement showing the sampling moments in relation to hydrological cycles. In all, graphs in the supplement had better quality. Authors could rewrite the text considering clarity. I think there is space for further digestion of the data because the main document contained 13 figures and the supplement had an additional 23 figures. At a quick glance, some figures in the supplement could be relevant/better in the main document but I recommend authors to re-think how to present the data with reasonable number of figures and tables (<10). 

As previously mentioned, incorporating diverse and heterogeneous data into the manuscript posed a considerable challenge. Determining which figures/tables belonged in the main manuscript versus the supplementary information proved to be a complex task. In response to the reviewer's guidance, the revised version of the manuscript features a rearrangement of information between the main manuscript and supplementary materials, accompanied by the removal of some figures (refer to the table provided in the response letter for details). Despite these adjustments, the supplementary information in the new version of the manuscript still contains a substantial number of figures (22) and tables (5). This is a consequence of the varying types of data employed. Notably, the profiles occupy multiple spots of the supplementary information. For instance, a single figure encompasses information on the vertical behavior in the water column of temperature, oxygen, turbidity, pH, and conductivity for 5 sampling points (comprising 3 locations, and morning, afternoon, and night for one of them) in a specific campaign.

Reviewing would have been much smoother if the figures had been in an order and accompanied the legend text.

The inclusion of figures in-line with the text is not permissible according to the journal's guidelines. In the figure file, all figures are presented in order and accompanied by the respective caption.

L 22. Why surface in the title and here? Concentrations and physicochemical parameters were studied across the water column.

In response to this comment, we have changed the title by: “CO2 and CH4 dynamics in a eutrophic tropical Andean reservoir”

L 63. dissolved what?

Dissolved gases. Corrected.

L. 77. or thereabouts. Could you formulate hypotheses?

We added the following sentence at the end of the introduction section: “We hypothesized that the surface behavior of greenhouse gases within a eutrophic reservoir suffer seasonal variations in response to changes in hydrological and meteorological conditions and that the spatial distribution of these gases varies with proximity to the river inflow” (L 69-72).

L. 111. water quality is related to the hydrological status?

Yes. It is expected that the hydrological status of the catchment is related to the water quality in the reservoir, either by causing pollutant dilution or by enriching the water body with substances transported by runoff. nevertheless, the statement has been removed from the manuscript in the revised version.

L. 120. Clarify the diurnal sampling. Where, how frequently etc.

We added the following sentence: “To study daily patterns, we conducted sampling at the dam zone (sampling point P1) three times a day during: morning (P1-M, 08:00-12:00), afternoon (P1-A, 14:00-20:00), and night (P1-N, 22:00-03:00). Typically, sampling followed a sequence: inflow zone P3 during the daytime, mid-lake zone P2 during the daytime, and finally, the dam zone (P1), capturing diurnal dynamics in all field campaigns, except the initial one (C1-H-Wet)” (L. 128-132). 

L. 153. How stable the water layers were?

The vertical structure was generally stable over time. To be clearer, we added the following information: “The stratification of the reservoir showed a persistent spatial pattern with distinct layers developing from the inflow zone (P3) towards the dam zone (P1). In the inflow region (~ 24 m water depth), there is a dominant inflow plume and small thermocline and surface layer, with a thin bottom layer appearing only in a few cases. In the mid-lake zone (~ 53 m depth) the inflow plume became thicker, but the observed vertical structure from the inflow zone persisted. In the lacustrine dam zone (~ 112 m depth), an anoxic bottom layer developed, except during the medium-level-wet campaign in Nov/2018 (C5-M-wet), when the cold inflow plume propagated along the bottom up to the dam (S4 Fig).” (L 281-288)

L. 176. Headspace technique …and concentrations were analyzed using a GC

Yes, they were measured using a GC. The section “Dissolved CO2 and CH4 in the water column” was completed re-written, even following the correction of the reviewer (L 182-194).

L. 228. How long is the chamber closure? In total 5 samples per closure?

The section “Air-water fluxes of CO2 and CH4, and 𝑘600” was partly re-written following the suggestion of the reviewer (L 218-221). The text now includes: “Fluxes of CO2 and CH4 across the air-water interface were measured using floating chambers. Two plastic chambers (volume 40 L, surface 0.15 m2), each equipped with a rubber stopper allowing for gas sampling with a syringe and needle, were deployed simultaneously from a boat. Each deployment lasted for 45 min and gas samples were collected in 15 min intervals”.

L. 232. How were the vials prepared? Vacuumed? Flushed with N2? Was the amount sufficient for the GC?

The section “Air-water fluxes of CO2 and CH4, and 𝑘600” was partly re-written following the information suggested by the reviewer. The text now includes: “After collection, samples were immediately stored in 20 mL vials previously sealed and vacuumed for CO2. For CH4 analysis, the vials were prepared with a KCl solution, while injecting the sample, the vials kept bottom-up while simultaneously expelling 10 mL of liquid through a second needle. Subsequently, gas concentrations in the samples were analyzed using gas chromatography (Shimadzu GC-2014 equipped with a methanizer and a flame ionization detector)” (L 222-225).

L 241. Why this threshold value?

The data obtained from floating chambers were affected by bubbles, as previously noted. We chose to include measurements associated with regressions displaying a reasonably high coefficient of determination (r2 > 0.70), as adopting a more stringent criterion would result in a reduction of available data points.

L. 511. could you show CH4 and CO2 fluxes in the same figure, with O2 and pH?

The oxygen and pH profiles, along with turbidity, temperature, and electrical conductivity, held relevance in delineating stratification and determining sampling depths for dissolved gases such as CO2 and CH4. Consequently, the coauthors find it sensible to maintain a distinction between the physicochemical profiles and those of dissolved gases. Nevertheless, the established stratification is visually represented by background colors in all profiles, delineating layers and aiding comprehension of the physicochemical characteristics in the dissolved gases profiles. 

L. 517. think how to average fluxes in the cases that diurnal dynamics. What’s the proportional effect of diurnal dynamics.

Given the scarcity of data points for diurnal variability, we opt not to attempt the estimation of the effect of diurnal dynamics in this study. 

L. 530. velocities based on measurements.

Owing to the length constraints of the original manuscript, we decided to focus the analysis on the water column and utilize the gas transfer velocity obtained from the floating chambers. The manuscript no longer includes the comparison of estimated gas transfer with previous models, and the use of the surface renewal model to propose a specific-site proportional constant has also been excluded.

L. 532. correlation with estimates based on measurements. Be precise, and explain what is acceptable correlation?

The data obtained from floating chambers were affected by bubbles, as previously noted. We chose to include measurements associated with regressions displaying a reasonably high coefficient of determination (r2 > 0.70), as adopting a more stringent criterion would result in a reduction of available data points. 

L. 616. balance was calculated. Estimated?

We have replaced “calculated” by “estimated”.

L. 633. estimated net CH4 production...

The caption of the figure (before Fig 13, now S21 Fig) was corrected according to the suggestion of the reviewer. 

L. 640. how was it observed? I suggest editing.

The section “Trophic state and CO2” was edited, according to the suggestions of the reviewer, now the text states: “Eutrophic conditions in the reservoir during the whole study period was indicated by the high phytoplankton abundance shown by visual observation of water color during the field campaigns, the recurrently observed negative CO2 fluxes (FCO2) during daytime, the supersaturation and large diel variations in dissolved oxygen (DO), the hypoxic to anoxic conditions in the hypolimnion (<10% of saturation), and the consistently low Secchi disk depth (SD < 2 m) at most of the sampling stations [27]. This agrees with the Toledo’s trophic state index (TSI > 54, company information) [28] (S22 Fig)” (L 527-553). 

L. 687. if not corrected with alkalinity and pH value?

The suggestion of the reviewer is on the following text: “It is widely known that dissolved CO2 is affected by the dynamic chemical equilibrium with other carbonate species”. As the reviewer points out, this could potentially be addressed by considering alkalinity and pH values. In fact, we address this concern a few lines later by stating, "It is possible to correct CO2 concentration estimates using pH and alkalinity, with better approximations in waters with neutral to high pH and high alkalinities (>1000 µmol/L); otherwise, overestimation (between 50 and 300%) can occur (Abril et al., 2015)”. However, this information is not included in the main manuscript as it is not a central point of our discussion but rather an explanation of our results. This paragraph is now part of the supplementary information section titled "On the potential overestimation of CO2 concentration."

L. 692. basic > alkaline?

We changed the word “basic” by “alkaline” following this suggestion.

L. 727. the shallower water body the higher CH4 fluxes, a well reported and typical pattern in lakes

We considered valuable to discuss our results with other studies in tropical reservoirs. Regarding this, we have added the following text: “Only a limited number of studies have investigated into the seasonal fluctuations of methane diffusive emissions from tropical reservoirs. It is not meaningful to compare the current findings with high-latitude systems, as their hydrological patterns differ significantly. Tropical systems experience distinct rainy and dry seasons, and the reservoirs maintain a weak stratification throughout the year. In contrast, temperate systems typically exhibit strong stratification in summer and complete mixing in autumn and winter, while boreal systems are iced-covered during winter”.

L 818. indirectly demonstrate? Did you measure microbial activity?

The reviewer's comment relates to the conclusions section, specifically the text: "Measurements in a eutrophic tropical reservoir indirectly indicate that the surface layer is a region of intense microbial activity, where alternating oxic production and oxidation likely play a significant role in the CH4 budget". Our intention here was to highlight, through mass balances, that the term of reactions, likely associated with microbial activity, held particular significance for the methane budget. However, as suggested by the reviewer, the original sentence might be misconstrued to imply direct measurement of microbial activity. To address this, we have rephrased this section (Lines 715-727).

L- 820 Diurnal only > what you mean? Sampling over a 24h cycle only versus over a year?

When we refer to "Diurnal only," we are essentially indicating "daytime." This sentence has been amended in response to the reviewer's feedback (L. 726).

---

## [Decision Letter · Decision Letter 1]

22 Jan 2024

CO2 and CH4 dynamics in a eutrophic tropical Andean reservoir

PONE-D-23-24316R1

Dear Dr. Lorke,

We’re pleased to inform you that your manuscript has been judged scientifically suitable for publication and will be formally accepted for publication once it meets all outstanding technical requirements.

Kind regards,

Steven Arthur Loiselle

Academic Editor

PLOS ONE

Reviewers' comments:

Reviewer's Responses to Questions

**Comments to the Author**

1. If the authors have adequately addressed your comments raised in a previous round of review and you feel that this manuscript is now acceptable for publication, you may indicate that here to bypass the “Comments to the Author” section, enter your conflict of interest statement in the “Confidential to Editor” section, and submit your "Accept" recommendation.

Reviewer #1: All comments have been addressed

2. Is the manuscript technically sound, and do the data support the conclusions?

Reviewer #1: Yes

3. Has the statistical analysis been performed appropriately and rigorously? 

Reviewer #1: Yes

4. Have the authors made all data underlying the findings in their manuscript fully available?

Reviewer #1: Yes

5. Is the manuscript presented in an intelligible fashion and written in standard English?

Reviewer #1: Yes

6. Review Comments to the Author

Reviewer #1: Relevant article and carefully reviewed by the authors. All suggestions were attended, I recommend publication in PLOS ONE.

7. PLOS authors have the option to publish the peer review history of their article (what does this mean?). If published, this will include your full peer review and any attached files.

Reviewer #1: No

---

## [Editor Report · Acceptance letter]

23 Feb 2024

PONE-D-23-24316R1 

PLOS ONE

Dear Dr. Lorke, 

I'm pleased to inform you that your manuscript has been deemed suitable for publication in PLOS ONE. Congratulations! Your manuscript is now being handed over to our production team.

Kind regards, 

on behalf of

Dr. Steven Arthur Loiselle 

Academic Editor

PLOS ONE